# TMEM55B links autophagy flux, lysosomal repair, and TFE3 activation in response to oxidative stress

Eutteum Jeong[1], Rose Willett[1], Alberto Rissone[1], Martina La Spina[1] & Rosa Puertollano [1] ✉

Lysosomes have emerged as critical regulators of cellular homeostasis. Here we show that the lysosomal protein TMEM55B contributes to restore cellular homeostasis in response to oxidative stress by three different mechanisms: (1) TMEM55B mediates NEDD4-dependent PLEKHM1 ubiquitination, causing PLEKHM1 proteasomal degradation and halting autophagosome/lysosome fusion; (2) TMEM55B promotes recruitment of components of the ESCRT machinery to lysosomal membranes to stimulate lysosomal repair; and (3) TMEM55B sequesters the FLCN/FNIP complex to facilitate translocation of the transcription factor TFE3 to the nucleus, allowing expression of transcriptional programs that enable cellular adaptation to stress. Knockout of *tmem55* genes in zebrafish embryos increases their susceptibility to oxidative stress, causing early death of *tmem55*-KO animals in response to arsenite toxicity. Altogether, our work identifies a role for TMEM55B as a molecular sensor that coordinates autophagosome degradation, lysosomal repair, and activation of stress responses.

Lysosomes are membrane-bound organelles found in all eukaryotic cells. They constitute the main degradative cellular compartment, enclosing a wide repertoire of acidic hydrolases capable of digesting macromolecules, including proteins, glycans, lipids and nucleic acids[1]. The catalytic function of lysosomes is essential for many cellular processes, such as turnover of cellular components, downregulation of surface receptors, bone remodeling, inactivation of pathogenic organisms, and antigen presentation.

Lysosomes have also recently emerged as unique signaling hubs[2]. Multiple types of signals are sensed and integrated at the lysosome-limiting membrane, allowing modulation of critical cellular processes, such as nutrient sensing, energy metabolism, immune response, and lysosomal damage repair[3]. The activation of these pathways requires the regulated recruitment of signaling complexes to the lysosomal surface, as well as the subsequent activation of specific transcription factors that modulate expression of elaborate transcriptional networks to facilitate cellular adaptation and maintain cellular homeostasis[4].

Equally important is the regulation of lysosome motility and positioning. The overall distribution of lysosomes is determined by the interaction of lysosomal proteins with microtubule motors and motor adapters, as well as by the establishment of physical contacts with other cellular compartments[5]. The bidirectional movement of lysosomes along microtubules is critical to facilitate autophagosome degradation, microbial killing, and antigen presentation, and is influenced by nutrient levels and cancer invasion[6].

A major open question in the field is understanding how processes as diverse as lysosomal biogenesis, repair, positioning, signaling and catabolism, can be simultaneously regulated in response to different types of stress. Our laboratory previously identified TMEM55B as an important regulator of lysosomal positioning. TMEM55B is a lysosomal transmembrane protein that binds to the dynein adapter JIP4, thus promoting dynein-dynactin-dependent lysosomal trafficking to the perinuclear region[7]. TMEM55B expression is regulated by the transcription factors TFEB and TFE3. Activation of TFEB/TFE3

[1]Cell and Developmental Biology Center, National Heart, Lung, and Blood Institute, National Institutes of Health, Bethesda, MD, USA.
✉e-mail: puertolr@mail.nih.gov

following starvation increases TMEM55B protein levels, leading to increased JIP4 recruitment to the lysosomal surface[7]. JIP4-mediated retrograde transport of lysosomes facilitates interaction and fusion between autophagosomes and lysosomes, resulting in degradation of autophagosomal content and restoration of energy homeostasis[7–9]. Therefore, the TFEB/TFE3-TMEM55B-JIP4 axis is critical to coordinate cellular adaptation to nutrient deprivation.

In this study we describe a role for TMEMB55B orchestrating cellular responses to lysosomal damage caused by acute oxidative stress. We found that the TMEM55B/JIP4 complex dissociates in response to sodium arsenite ($NaAsO_2$), allowing TMEM55B-dependent recruitment of HECT domain E3 ligases to the lysosomal surface. This results in ubiquitination and consequent proteasomal degradation of PLEKHM1, thus halting fusion of autophagosomes with potentially damaged lysosomes. Additionally, TMEM55B interacts with components of the ESCRT complex via a PSAP motif to facilitate lysosomal repair, and sequesters folliculin, causing TFE3 activation. These results unveil an unexpected link between catabolism, lysosomal repair, and transcriptional response to oxidative stress.

## Results

### TMEM55B interacts with NEDD4-like E3 ligases

TMEM55B is a transmembrane protein that mainly localizes to lysosomes. Secondary structure prediction suggests that it is comprised of a large N-terminal cytosolic domain (CD), two transmembrane domains (TM), and a short C-terminal cytosolic tail (Fig. 1a). We performed immunoprecipitation of TMEM55B followed by mass spectrometry (MS) analysis with the goal of identifying TMEM55B binding partners. As expected, we confirmed the previously described interaction between TMEM55B and JIP4 (Supplementary Fig. 1a and Supplementary Data 1)[7]. Our MS analysis also pinpointed to several members of the C2-WW-HECT (or NEDD4-like) family of E3 ligases as potential interactors, including NEDD4, NEDD4L, WWP1, WWP2, SMURF1, SMURF2, and ITCH (Supplementary Fig. 1a). NEDD4-like E3 ligases consist of an N-terminal C2 calcium-binding domain, several WW domains, and a C-terminal HECT ubiquitin ligase domain (Supplementary Fig. 1b)[10]. The WW domains determine ligase specificity by interacting with PPXY (X denotes any amino acid in this position) motifs in target proteins.

Sequence analysis revealed that TMEM55B harbors a classical PPXY motif that could mediate direct binding to NEDD4-like ubiquitin ligases. This motif is highly conserved among species, being present not only in mammals but also in fish and fly (Fig. 1b). To test this, recombinant TMEM55B full length, as well as its cytosolic N-terminal domain, were immunoprecipitated via their GFP tags, and co-immunoprecipitation with endogenous NEDD4 was evaluated. As seen in Fig. 1c, both TMEM55B full length (FL) and TMEM55B-CD (CD) efficiently immunoprecipitated NEDD4. To confirm that the binding is dependent on the PPXY motif, we constructed a mutant in which proline 66 was mutated to alanine (P66A). This mutation completely abolished the ability of TMEM55B to interact with NEDD4, confirming that TMEM55B binding to NEDD4 occurs via PPXY motifs. In contrast, mutation of P66 did not affect the binding of TMEM55B to JIP4, suggesting that this mutation does not alter the conformation of the protein (Fig. 1c).

To verify the interaction between TMEM55B and NEDD4, we transfected U2OS cells stably expressing NEDD4-Cherry with either GFP or TMEM55B-GFP. As seen in Fig. 1d, NEDD4 showed a diffuse cytosolic distribution in GFP-expressing cells, but it was recruited to lysosomes when co-expressed with TMEM55B-GFP. Consistent with the requirement of the PPXY motif for the binding between TMEM55B and NEDD4, we did not observe recruitment of NEDD4 to lysosomes upon expression of the TMEM55B-P66A-GFP mutant (Fig. 1d). It is important to note that both, TMEM55B-GFP and TMEM55B-P66A-GFP were efficiently transported to lysosomes, indicating that NEDD4 is not

required for TMEM55B sorting. PPXY-dependent interaction of TMEM55B-GFP with other members of the NEDD4-like family of ubiquitin ligases, including NEDD4L and ITCH, was confirmed by co-immunoprecipitation (Fig. 1e).

Next, we assessed whether the interaction of TMEM55B with NEDD4-like E3 ligases might be influenced by stress. For that we treated U2OS cells expressing TMEM55B-GFP with different stressors, including starvation (EBSS), oxidative stress ($NaAsO_2$ and $H_2O_2$), lysosomal damaging agents (LLOMe), mitochondrial uncoupling compounds (CCCP), and inductors of endoplasmic reticulum stress (tunicamycin and thapsigargin). In agreement with our previous study[7], we observed a shift in TMEM55B electrophoretic mobility in response to $NaAsO_2$, which likely corresponds to increased phosphorylation (Fig. 1f). Interestingly, we found that the binding of NEDD4L to TMEM55-GFP was increased under oxidative stress conditions (Fig. 1f and Supplementary Fig. 1c). Similar to recombinant TMEM55B-GFP, endogenous TMEM55B localized to LAMP1-positive structures and appeared in clustered lysosomes upon $NaAsO_2$ treatment (Supplementary Fig. 1d). Endogenous TMEM55B also pulled-down NEDD4L with a higher efficiency under oxidative stress conditions (Supplementary Fig. 1e, f). As expected, the interaction between TMEM55B-GFP and NEDD4L in $NaAsO_2$ conditions was still abolished by mutation of the PPXY motif (Supplementary Fig. 1g, h).

The ligase activity of NEDD4L is tightly regulated by autoinhibition. Binding of the WW domain locks the HECT domain in an inactive state by inhibiting E2-E3 transthiolation[11]. Phosphorylation of NEDD4L in serine 448 by different kinases, including SGK1, PKA and AKT, contribute to its inhibition[12]. Under stress conditions, such as starvation, decrease in phosphorylation of NEDD4L results in HECT oligomerization and activation[13]. In agreement with these studies, we observed a significant reduction in NEDD4L S448 phosphorylation when cells were incubated in starvation media (EBSS) for 4 h (Supplementary Fig. 1i, j). Reduced NEDD4L phosphorylation, and thereby activation, was also observed when oxidative stress was induced by treatment with $NaAsO_2$ (Supplementary Fig. 1i, j). Furthermore, phosphorylation of ITCH at T222, a modification known to enhance its catalytic activity[14], was observed following incubation with $NaAsO_2$, indicating that oxidative stress has a broad effect in the activation of NEDD4-like E3 ligases (Supplementary Fig. 1k, l).

Altogether, our data suggest that TMEM55B recruits NEDD4-like E3 ligases to lysosomes, which upon stress-mediated activation, might mediate ubiquitination of lysosomal proteins.

### TMEM55B is ubiquitinated in response to oxidative stress

Next, we assessed whether TMEM55B was ubiquitinated in response to oxidative stress. As we have previously described, expression of recombinant TMEM55B induced lysosomal clustering in the perinuclear region (Fig. 1g)[7]. However, under these conditions, no noticeable accumulation of ubiquitin was observed on the lysosomal surface, which showed instead a predominant nuclear distribution (Fig. 1g). In contrast, generation of oxidative stress by treatment with $NaAsO_2$, a condition that activates NEDD4-like E3 ligases (Supplementary Fig. 1i, k), resulted in a clear concentration of ubiquitin on lysosomes, as determined by its co-localization with TMEM55B-GFP and the lysosomal marker LAMTOR1 (Fig. 1g). Accumulation of ubiquitin in the perinuclear region following oxidative stress was much less noticeable in cells expressing TMEM55B-P66A-GFP (Fig. 1g and Supplementary Fig. 1m), suggesting that the interaction of TMEM55B with NEDD4-like E3 ligases is required for the ubiquitination of lysosomal proteins in response to $NaAsO_2$.

To directly monitor TMEM55B ubiquitination, we expressed either TMEM55B-WT or TMEM55B-P66A, together with a plasmid encoding HA-tagged ubiquitin, in U2OS cells. Immunoprecipitation of TMEM55B under denaturing conditions followed by immunoblotting for ubiquitin revealed a high molecular weight smear in cells expressing

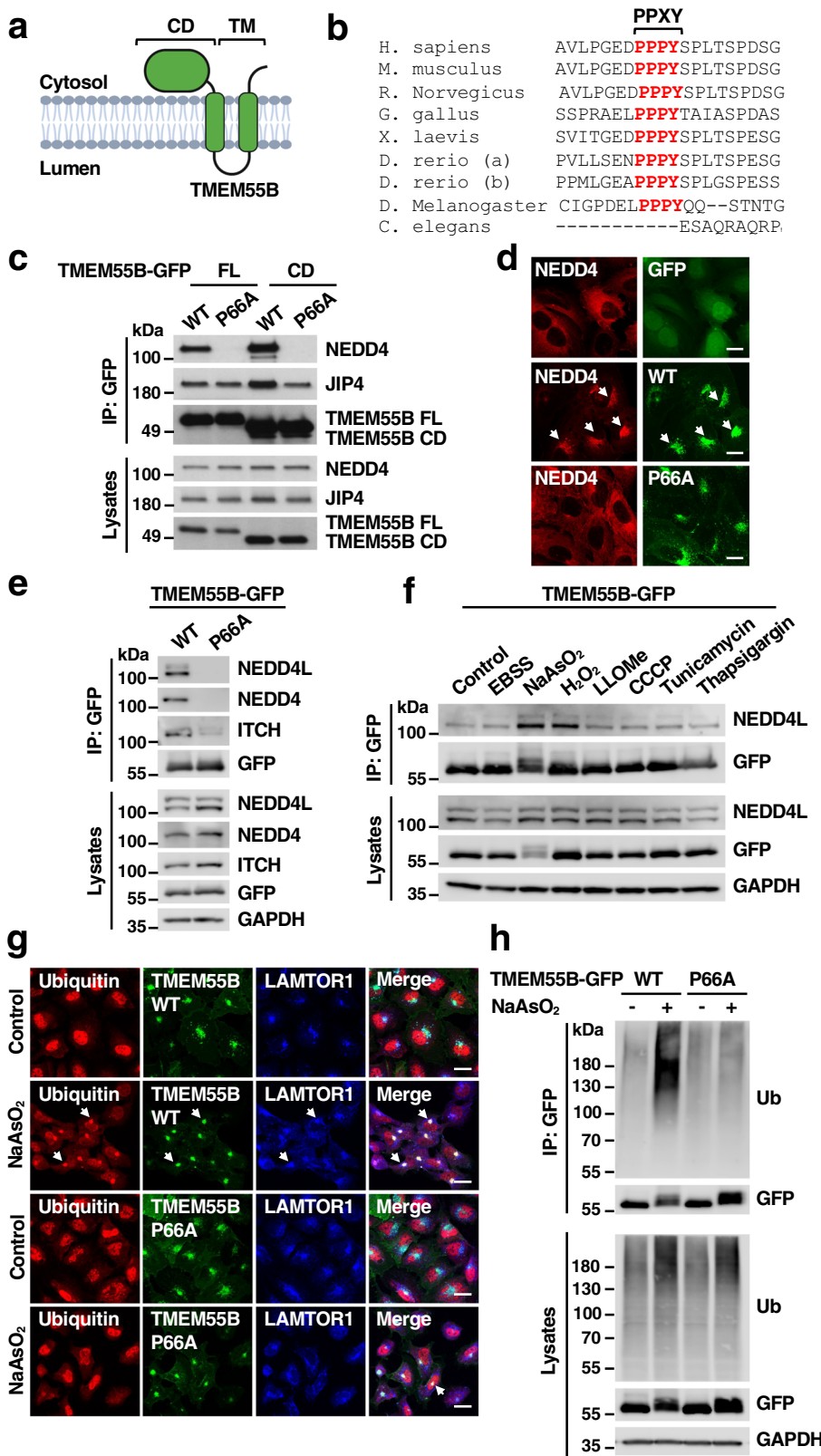

TMEM55B-WT-GFP upon treatment with NaAsO$_2$ (Fig. 1h). In contrast, TMEM55B-GFP ubiquitination was very modest under control conditions or upon mutation of the PPXY motif (Fig. 1h), suggesting that TMEM55B is ubiquitinated by NEDD4-like E3 ligases in response to oxidative stress.

To further confirm these results, we performed proteomic analysis. Cells expressing either TMEM55B-WT or TMEM55B-P66A were

treated with NaAsO$_2$ for 2 h. TMEM55B immunoprecipitates were then subjected to trypsin digestion, followed by purification of ubiquitinated peptides with anti-diGly antibodies and MS identification. Using this approach, seven ubiquitination sites were identified: K96, K103, K114, K120, K121, K134, and K148, all of them located in the CD domain. Importantly, we barely detected ubiquitinated peptides in samples isolated from unstimulated cells or from cells expressing the

**Fig. 1 | TMEM55B interacts with members of the NEDD4-like family of E3 ligases. a** Schematic of the predicted TMEM55B membrane topology. Illustration created with BioRender.com. **b** Multi-sequence alignment of TMEM55B orthologs in different species. The PPXY motif is indicated in red. **c** U2OS cells were transfected with plasmids encoding TMEM55B-GFP-WT full length (FL), TMEM55B-GFP-WT cytosolic domain (CD), TMEM55B-GFP-P66A FL, or P66A CD. Cells were lysed and pulled down with GFP beads. The results are representative of three independent experiments. WT (Wild-type), P66A (P66A mutant). **d** U2OS TMEM55B KO cells stably expressing mCherry-NEDD4 (red) were transfected with plasmids encoding GFP, TMEM55B-GFP-WT or P66A (green). Cells were fixed and permeabilized for immunofluorescence. Scale bars, 20 µm, $n = 3$. **e** U2OS cells were infected with adenovirus expressing TMEM55B-GFP-WT or P66A and pulled down with GFP beads. The results are representative of three independent experiments. **f** U2OS

cells infected with adenovirus expressing TMEM55B-GFP-WT were treated with various drugs and pulled down with GFP beads. EBSS for 4 h, NaAsO$_2$ (300 µM) for 2 h, H$_2$O$_2$ (500 µM) for 4 h, LLOMe (1 mM) for 2 h, CCCP (25 µM) for 4 h, Tunicamycin (10 µg/ml) for 4 h, Thapsigargin (10 µM) for 4 h. The results are representative of three independent experiments. **g** U2OS cells infected with adenovirus expressing TMEM55B-GFP-WT or P66A (green) were treated with or without NaAsO$_2$ (300 µM) for 2 h. Cells were fixed and immunostained with antibodies against ubiquitin (red) and LAMTOR1 (blue). Scale bars, 20 µm, $n = 3$. **h** U2OS cells were transfected with plasmids encoding TMEM55B-GFP-WT or P66A and treated with or without NaAsO$_2$ (300 µM) for 2 h. Cells were lysed and immunoprecipitated with GFP beads under denaturing condition. The results are representative of three independent experiments. Source data are provided as a Source Data file.

TMEM55B-P66A mutant (Supplementary Data 2). Altogether, our data indicate that TMEM55B is ubiquitinated in several residues in response to oxidative stress and that the PPXY motif is required for TMEM55B ubiquitination.

## TMEM55B interacts with PLEKHM1 under oxidative stress

We hypothesized that TMEM55B may not only recruit NEDD4L to lysosomes, but also NEDD4L targets, thus facilitating their ubiquitination. To test this, we first immunoprecipitated TMEM55B-GFP from U2OS cells treated with or without NaAsO$_2$ and used MS to identify proteins that display increased binding following stress. Interestingly, we found that PLEKHM1 (Pleckstrin homology domain family M member 1), as well as all the subunits of the HOPS (homotypic fusion and vacuole protein sorting) complex (VPS39, VPS41, VPS11, VPS16, VPS18 and VPS33), showed increased binding to TMEM55B under oxidative stress conditions (Fig. 2a and Supplementary Data 1).

PLEKHM1 is an adapter protein that binds Rab7, the HOPS complex, and LC3/GABARAP to facilitate fusion between autophagosomes and lysosomes[15]. To confirm our MS results, we performed co-immunoprecipitation experiments. As shown in Fig. 2b, c, endogenous TMEM55B pulled-down PLEKHM1 and the HOPS subunit VPS41, but not LAMP1, following NaAsO$_2$ treatment. The interaction seemed to be very specific for NaAsO$_2$, as we did not detect binding of PLEKHM1 or VPS41 to either recombinant or endogenous TMEM55B when cells were incubated with a variety of stressors, including EBSS, H$_2$O$_2$, CCCP, LLOMe, tunicamycin or thapsigargin (Fig. 2d and Supplementary Fig. 2a). Very unexpected was the observation that the interaction between TMEM55B and JIP4 was disrupted by NaAsO$_2$, suggesting that the binding of TMEM55B to either JIP4 or PLEKHM1 is mutually exclusive (Fig. 2c, d, and Supplementary Fig. 2a). Similar results were obtained in HeLa cells expressing recombinant TMEM55B-GFP (Fig. 2e).

The interaction was further confirmed by reverse immunoprecipitation experiments in which PLEKHM1-Flag was pulled-down under control or NaAsO$_2$ conditions. We observed that even though PLEKHM1-Flag and TMEM55B-GFP co-localize to Rab7-positive puncta both in control and NaAsO$_2$-treated cells (Supplementary Fig. 2b), the interaction between the two proteins was only detected upon incubation with NaAsO$_2$ (Supplementary Fig. 2c, d).

Next, we sought to identify which PLEKHM1 domain is implicated in the binding to TMEM55B. PLEKHM1 contains an N-terminal RUN (RUNDC3A/RPIP8, UNC-14 and RUSC1/NESCA) domain, two internal PH (Pleckstrin homology) domains, and a C-terminal C1 domain/zinc-finger-like motif (Fig. 2f). It has been reported that the RUN domain binds HOPS, while the PH2 and C1 domains mediate the interaction with active Rab7. In addition, a LIR (LC3-interacting) motif located between the PH1 and PH2 domains mediates binding to LC3/GABARAP[16]. Therefore, we generated several chimeras lacking different PLEKHM1 domains (Fig. 2f) and tested their ability to interact with both endogenous and recombinant TMEM55B. As seen in Fig. 2g and Supplementary Fig. 2e, only those chimeras containing the PH1

domain (residues 533-625) were able to bind TMEM55B following NaAsO$_2$ treatment, suggesting that the LIR (residues 632-638), PH2 (residues 683-777), and C1 (residues 986-1040) domains are dispensable for the interaction. We also observed that recombinant PLEKHM1-Flag co-immunoprecipitated TMEM55-GFP full length (FL), as well as its N-terminal cytosolic domain (CD) (Supplementary Fig. 2f). Altogether, our results identify a role for the PH1 domain of PLEKHM1 in mediating binding to TMEM55B under oxidative stress conditions.

## Acute oxidative stress induces TMEM55B phosphorylation

Next, we question why NaAsO$_2$ was the only tested agent capable of causing JIP4 dissociation and PLEKHM1/VPS41 binding. One possible explanation is that only NaAsO$_2$ changed TMEM55B electrophoretic mobility, suggesting that this compound may induce TMEM55B phosphorylation. To test this possibility, lysates and TMEM55B immunoprecipitates obtained from NaAsO$_2$-treated cells were incubated with or without Lambda phosphatase. As predicted, the TMEM55B higher molecular weight band observed upon NaAsO$_2$ treatment almost disappeared after incubation with Lambda phosphatase, strongly indicating that this modification does indeed correspond to phosphorylation (Fig. 3a). Quantification of three independent experiments is shown in Fig. 3b. Furthermore, mass spectrometry analysis identified two TMEM55B residues, T111 and S162, which undergo phosphorylation in response to NaAsO$_2$ (Supplementary Data 3 and 4). The percentage of peptides phosphorylated at S162 increased from 6.98% to almost 30% in NaAsO$_2$-treated cells, suggesting that this residue may be key for TMEM55B regulation (Supplementary Fig. 3a). We then assessed whether TMEM55B phosphorylation occurs in response to other agents known to cause oxidative stress. Interestingly, we found that those compounds that generated high ROS levels, including NaAsO$_2$ and acrolein, also caused TMEM55B phosphorylation and TMEM55B/JIP4 dissociation (Fig. 3c, Supplementary Fig. 3b, c). Conversely, compounds that were less efficient ROS inductors, such as H$_2$O$_2$ and spermidine, failed to induce TMEM55B phosphorylation or JIP4 dissociation (Fig. 3c, Supplementary Fig. 3b, c). These results further demonstrate a role of TMEM55B in cellular response to acute oxidative stress.

Recent evidence has suggested that PLEKHM1 is directly phosphorylated by mTOR and Erk2 MAPK[17] at Ser432/Ser435, while TMEM55B undergoes Erk-dependent phosphorylation in RAW 264.7 macrophages treated with Toll-like receptor ligands[8]. In addition, it is well-established that p38, JNK and Erk1/2 MAPK get strongly activated in response to oxidative stress[18]. Therefore, we investigated whether these kinases regulate TMEM55B phosphorylation in response to acute oxidative stress. To test this, cells were treated with NaAsO$_2$ in the presence of different kinase inhibitors specific for p38 MAPK (SB203580), JNK (JNK Inhibitor VIII), Erk1/2 (U0126), and mTOR (Torin-1). The levels of phospho-p38, phospho-JNK, phospho-Erk1/2, as well as the mTOR target phospho-S6K, were measured as control for the efficiency of the inhibition (Supplementary Fig. 3d, e). However, none of the tested inhibitors prevented NaAsO$_2$-induced TMEM55B

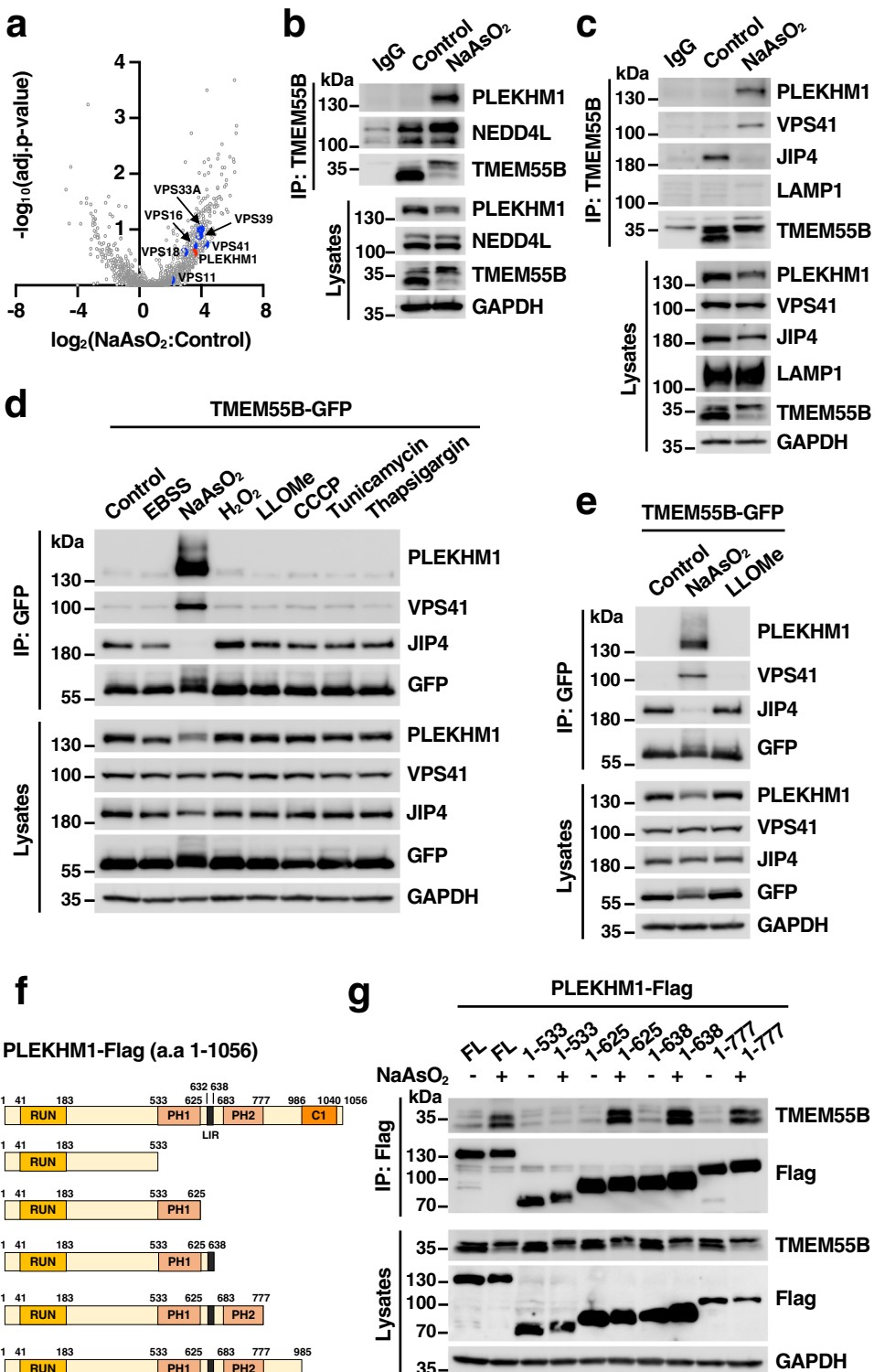

**Fig. 2 | PLEKHM1 interacts with TMEM55B upon NaAsO₂ treatment. a** Volcano plot of hits identified in immunoprecipitation and mass spectrometry analysis from U2OS cells infected with adenovirus expressing TMEM55B-GFP-WT treated with NaAsO₂ (300 μM), compared to untreated cells. The data were analyzed with two-tailed *t* test. **b, c** U2OS cells were treated with or without NaAsO₂ (300 μM) for 2 h and immunoprecipitated with anti-TMEM55B antibody. The results are representative of three independent experiments. **d** U2OS cells infected with adenovirus expressing TMEM55B-GFP-WT were treated with various drugs and pulled down with GFP beads. EBSS for 4 h, NaAsO₂ (300 μM) for 2 h, H₂O₂ (500 μM) for 4 h, LLOMe (1 mM) for 2 h, CCCP (25 μM) for 4 h, Tunicamycin (10 μg/ml) for 4 h,

Thapsigargin (10 μM) for 4 h. The results are representative of three independent experiments. **e** Hela cells infected with adenovirus expressing TMEM55B-GFP were treated with NaAsO₂ (300 μM) or LLOMe (1 mM) for 2 h and pulled down with GFP beads. The results are representative of three independent experiments. **f** Schematic representation of the PLEKHM1 and truncated mutants. **g** U2OS cells were transfected with Flag-tagged PLEKHM1 plasmid variants and treated with or without NaAsO₂ (300 μM) for 2 h. Cells were lysed and immunoprecipitated with Flag beads. The results are representative of two independent experiments. Source data are provided as a Source Data file.

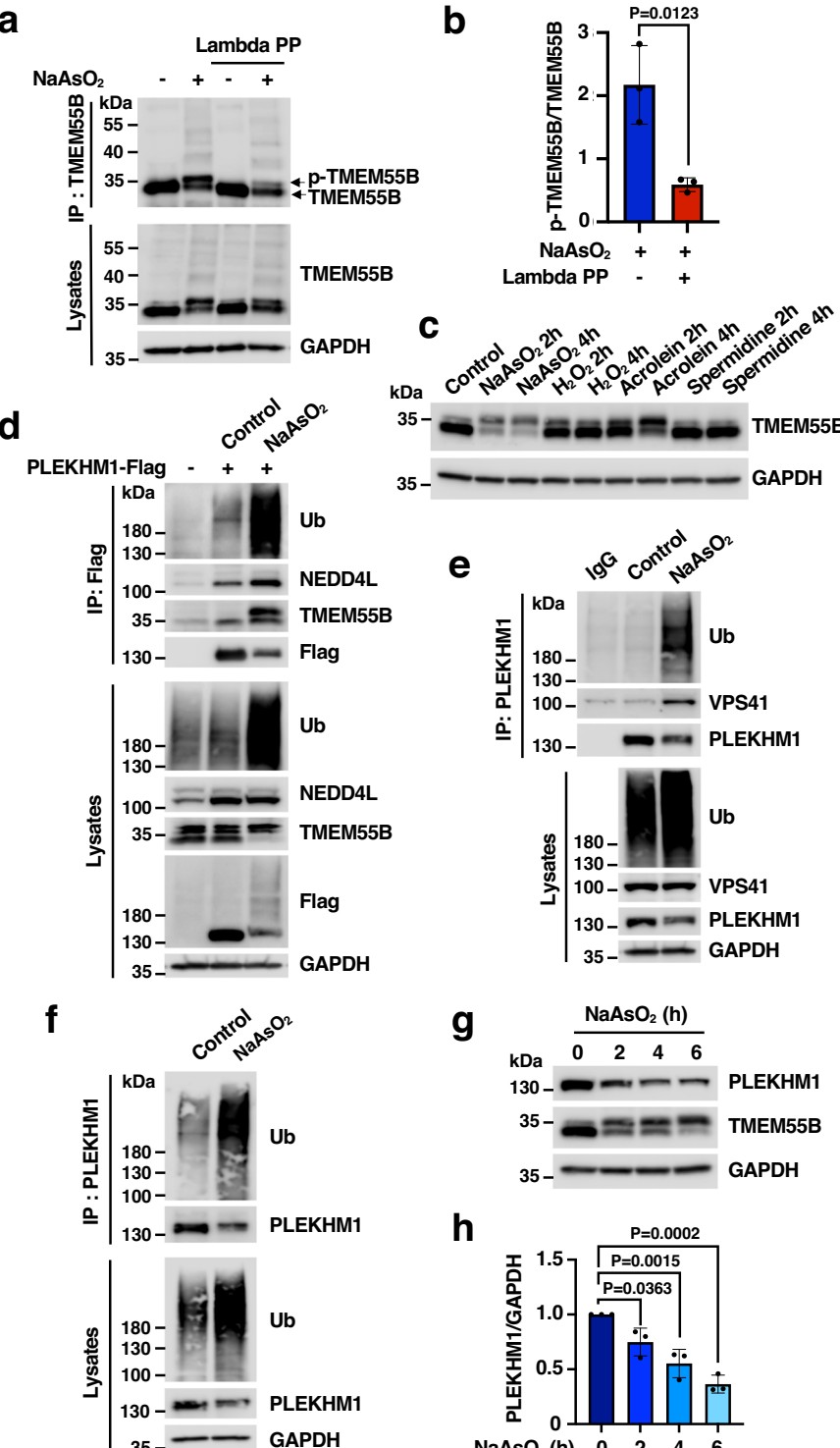

**Fig. 3 | NaAsO₂ induces TMEM55B phosphorylation and PLEKHM1 ubiquitination.** a U2OS cells were treated with or without NaAsO₂ (300 μM) for 2 h. Cell lysates and samples immunoprecipitated with anti-TMEM55B antibody were incubated with or without Lambda phosphatase. The results are representative of three independent experiments. **b** Quantification of immunoblots shown in (**a**). The data represent means ± SEM, *n* = 3 independent experiments. Statistical significance was determined by using two-tailed unpaired *t* test. **c** U2OS cells were treated with NaAsO₂ (300 μM), H₂O₂ (500 μM), Acrolein (200 μM) or Spermidine (300 μM) as indicated times. The results are representative of two independent experiments. **d** U2OS cells transfected with plasmids encoding PLEKHM1-Flag were treated with or without NaAsO₂ (300 μM) for 2 h and immunoprecipitated with Flag beads. The results are representative of three independent experiments. **e** U2OS cells were

treated with or without NaAsO₂ (300 μM) for 2 h and immunoprecipitated with anti-PLEKHM1 antibody. The results are representative of two independent experiments. **f** U2OS cells were treated with NaAsO₂ (300 μM) for 2 h and immunoprecipitated with anti-PLEKHM1 antibody under denaturing conditions. The results are representative of three independent experiments. **g** U2OS cells were treated with NaAsO₂ (300 μM) for indicated times and analyzes by immunoblot with the indicated antibodies. The results are representative of three independent experiments. **h** Quantification of immunoblots shown in (**g**). The data represent means ± SEM, *n* = 3 independent experiments. Statistical significance was determined by using one-way ANOVA with Dunnett's multiple comparisons. Source data are provided as a Source Data file.

phosphorylation, nor interaction of TMEM55B with PLEKHM1 and VPS41, or JIP4 detachment (Supplementary Fig. 3d, e). Furthermore, JAK3 IV inhibitors, which were recently reported to block oxidative stress-mediated phosphorylation of JIP4[9], did not stop the dissociation of the TMEM55B/JIP4 complex under our experimental conditions (Supplementary Fig. 3d). These results indicate that mTOR and MAPKs do not play a major role in regulating TMEM55B/PLEKHM1 interaction.

**TMEM55B facilitates PLEKHM1 proteasomal degradation**

Next, we addressed whether the interaction of PLEKHM1 with TMEM55B may promote its ubiquitination. We found that not only the interaction of PLEKHM1-Flag with endogenous TMEM55B and NEDD4L increased with NaAsO$_2$, but also its ubiquitination, as assessed by blotting immunoprecipitated PLEKHM1-Flag with antibodies against ubiquitin and by the smeared bands detected by the Flag antibody in total lysates (Fig. 3d). Ubiquitination of endogenous PLEKHM1 was also detected upon NaAsO$_2$ treatment (Fig. 3e, f). Importantly, a significant reduction in the total levels of recombinant and endogenous PLEKHM1 was observed after incubation with NaAsO$_2$ for 2 h, suggesting that PLEKHM1 ubiquitination may result in degradation of the protein (Fig. 3d–f). To further test this idea, we incubated U2OS cells with NaAsO$_2$ for 2 h, 4 h, and 6 h, and confirmed a progressive and significant reduction in PLEKHM1 protein levels (Fig. 3g, h).

We then compared PLEKHM1 levels in U2OS cells infected with adenovirus expressing either TMEM55B-GFP wild-type (WT) or the TMEM55B-GFP-P66A mutant (P66A) (Fig. 4a, b). Notably, whereas both TMEM55B WT and P66A showed strong binding to PLEKHM1 upon incubation with NaAsO$_2$, only TMEM55B WT recruited also NEDD4L (Fig. 4a). As a result, ubiquitination of PLELHM1 was only detected in cells expressing TMEM55B-WT (Fig. 4a). Furthermore, the NaAsO$_2$-induced reduction in PLEKHM1 protein levels was less noticeable in cell expressing the TMEM55B P66A mutant (Fig. 4a). Importantly, treatment with TAK243, a selective ubiquitin activating enzyme (UBA1) inhibitor, did not affect binding of TMEM55B to PLEKHM1, but prevented NaAsO$_2$-induced PLEKHM1 degradation, further demonstrating that this is a ubiquitination-mediated process (Supplementary Fig. 4a, b). Our observations were further confirmed by reversed immuno-precipitation experiments. PLEKHM1 pulled down both TMEM55B WT and the P66A mutant, but the binding to NEDD4L and the reduction in PLEKHM1 levels were much more pronounced upon TMEM55B-WT expression (Fig. 4b). Similar results were obtained in HeLa cells (Supplementary Fig. 4c). These results indicate that the ability of TMEM55B to recruit NEDD4L is critical for PLEKHM1 ubiquitination and degradation under oxidative stress.

To better understand the mechanism of PLEKHM1 degradation, we treated U2OS cells expressing either TMEM55B WT or P66A with NaAsO$_2$ for different periods of time. As expected, PLEKHM1 degradation was significantly stronger in cells expressing TMEM55B WT (Fig. 4c, d). Furthermore, PLEKHM1 degradation was blocked by incubation with the proteosome inhibitor MG132, but not by inhibiting lysosomal activity with bafilomycin A1, indicating that TMEM55B promotes PLEKHM1 ubiquitination and proteasomal degradation in response to NaAsO$_2$ (Fig. 4c, d). These results were further supported by the significantly reduced PLEKHM1 degradation observed in TMEM55B-depleted cells, while depletion of the family member TMEM55A had no effect (Fig. 4e, f).

It was previously reported that depletion of PLEKHM1 significantly reduces the fusion rate between autophagosomes and lysosomes[16]. We hypothesized that TMEM55B-mediated PLEKHM1 degradation may serve as a mechanism to slow down the delivery of degradative cargo under conditions of acute stress in which lysosomes may undergo damage or present reduced activity. In agreement with this idea, we found that the reduction in PLEKHM1 levels observed in TMEM55B-WT-expressing cells was accompanied by a decrease in autophagy flux, as assessed by the lack of increase in LC3$_{II}$ levels in cells treated with

bafilomycin A1. Conversely, the levels of PLEKHM1 remained constant in cells expressing TMEM55B P66A, resulting in continuous autophagosome/lysosome fusion (Fig. 4g, h).

**TMEM55B promotes ESCRT recruitment to lysosomes**

Next, we performed proteomic comparison between TMEM55B WT and P66A interactors following treatment with NaAsO$_2$. As expected, the P66A mutant displayed decreased binding to several NEDD4-like ligases, including NEDD4, NEDD4L, WWP1, WWP2, and ITCH (Fig. 5a and Supplementary Data 1). Interestingly, we also found reduced P66A affinity for several ubiquitin-binding proteins and ESCRT complex subunits. These include components of the ESCRT-0 (STAM2 and HRS), ESCRT-I (VPS37 and MVB12), and ESCRT-III (CHPM4B and CHMP5) sub-complexes, as well as the ESCRT regulatory proteins VPS4 and ALIX (Fig. 5a and Supplementary Data 1). Since the ESCRT complex plays an important role in mediating lysosome repair[19], we hypothesized that TMEM55B may facilitate ubiquitin-dependent ESCRT recruitment to the lysosomal membrane in response to oxidative stress. It is well established that the ESCRT complex interacts with proteins carrying P(S/T)AP and/or PPXY motifs. The P(S/T)AP domain binds to TSG101, which in turn binds ALIX[20], while recruitment of NEDD4-E3 ligases via PPXY motifs, and subsequent protein ubiquitination, further facilitates the recruitment of the ESCRT machinery[21]. Sequence analysis revealed that in addition to the PPXY motif, TMEM55B also contain a P(S/T)AP sequence in the N-terminal region (residues 40–43), and this sequence is highly conserved through evolution (Fig. 5b). Furthermore, the interaction between TMEM55B-GFP-WT and ESCRT proteins following NaAsO$_2$-treatment was confirmed by immunoprecipitation (Fig. 5c). To examine the relative contribution of the PPXY and P(S/T)AP motifs to ESCRT binding, we generated a new mutant in which serine 41 was substituted by alanine (S41A). As seen in Fig. 5c, the S41A mutant showed increased ubiquitination and NEDD4L binding in response to NaAsO$_2$. However, neither the S41A nor the P66A mutant, were able to interact with ESCRT proteins under stress. These results indicate that both the P(S/T)AP and PPXY motifs are necessary for efficient ESCRT recruitment to lysosomes. These observations were confirmed by immunofluorescence. Ubiquitination of lysosomal proteins upon NaAsO$_2$ treatment was observed in cells expressing either TMEM55-GFP-WT or the S41A mutant (Supplementary Fig. 5a). However, only TMEM55B-GFP-WT induced robust CHMP2B recruitment, further suggesting that TMEM55B ubiquitination alone is not sufficient to stimulate efficient ESCRT recruitment to lysosomes under oxidative stress conditions (Fig. 5d and Supplementary Fig. 5b). As expected, the P66A mutant failed to induce recruitment of both ubiquitin and ESCRT to lysosomes (Fig. 5d and Supplementary Fig. 5b).

ESCRT recruitment to lysosomes is an early event following lysosomal damage. Extensive damage that cannot be repaired by the ESCRT complex leads to accumulation of galectins, triggering elimination of damaged lysosomes by lysophagy. Consistent with the proposed role of TMEM55B in ESCRT recruitment to injured lysosomes, we observed significantly higher galectin3 accumulation in TMEM55B-depleted cells after 6 h of NaAsO$_2$ treatment (Fig. 5e, f). These results suggest that TMEM55B may halt fusion of autophagosomes with damaged lysosomes, while at the same time promoting their repair. However, it is important to note that the potential role of TMEM55B in lysosomal repair seems to be specific for oxidative stress, as the number of galectin3-positive puncta induced by LLOMe was not different between control (Null) and TMEM55B-depleted (TMEM55B-KO) cells (Supplementary Fig. 5c).

The ability of TMEM55B to facilitate ESCRT-mediated lysosome repair under oxidative stress conditions might provide a cell survival advantage. To test this hypothesis, we used flow cytometry to monitor cell death in Null and TMEM55B-KO U2OS cells after prolonged exposure to NaAsO$_2$. Both Null and TMEM55B-KO cells exhibited high

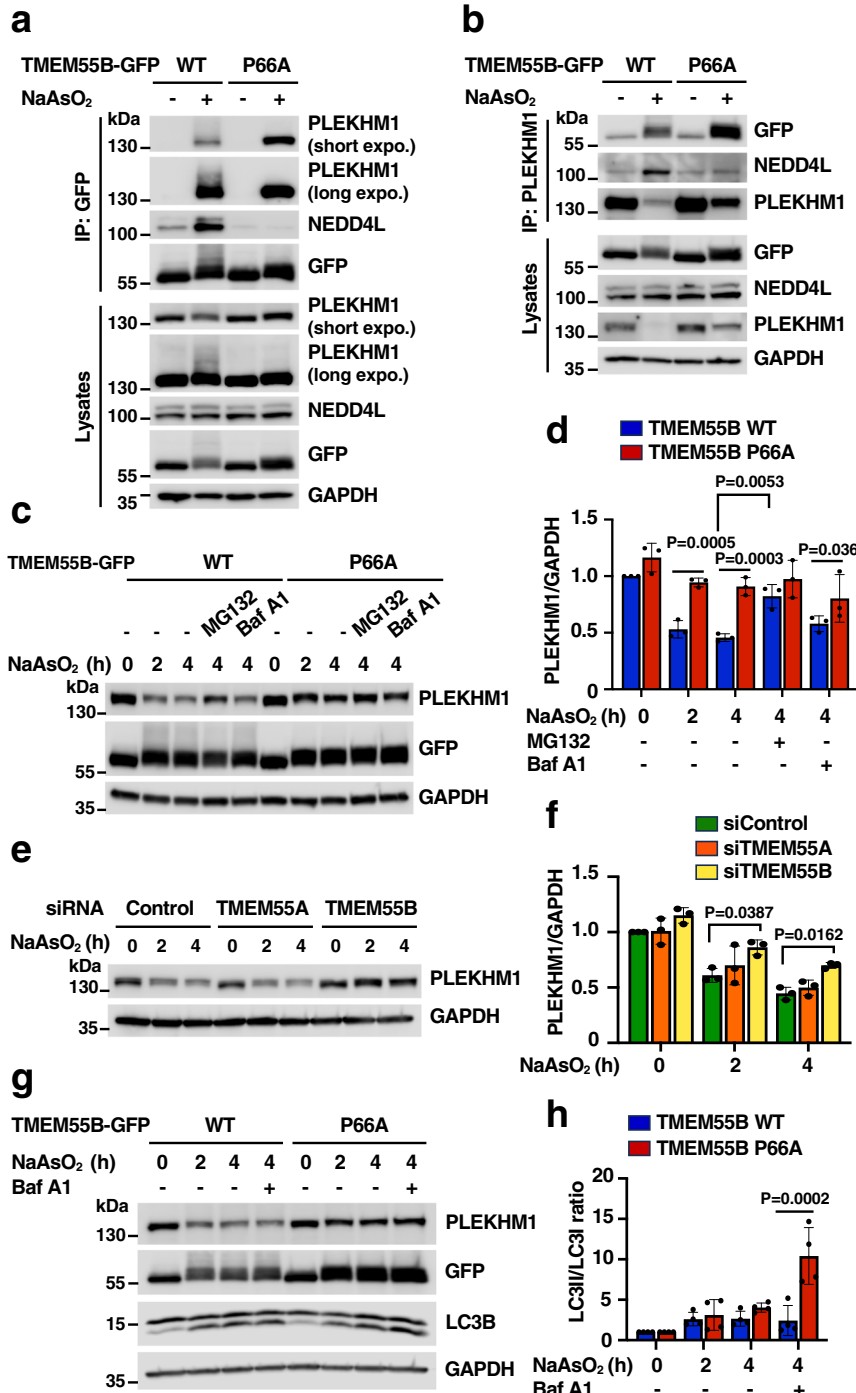

**Fig. 4 | TMEM55B-mediated PLEKHM1 degradation decreases autophagy flux.**
**a** U2OS cells infected with adenovirus expressing TMEM55B-GFP-WT or P66A were treated with or without NaAsO₂ (300 μM) for 2 h. Cells were lysed and pulled down with GFP beads. The results are representative of four independent experiments. Short expo. (Shorter exposure), long expo. (Longer exposure). **b** U2OS cells infected with adenovirus expressing TMEM55B-GFP-WT or P66A were treated with or without NaAsO₂ (300 μM) for 2 h. Cells were lysed and immunoprecipitated with anti-PLEKHM1 antibody. The results are representative of three independent experiments. **c** U2OS cells infected with adenovirus expressing TMEM55B-GFP-WT or P66A were incubated with NaAsO₂ (300 μM) in the presence of MG132 (50 μM) or Bafilomycin A1 (100 nM) for the indicated times. The results are representative of three independent experiments. **d** Quantification of immunoblots shown in (**c**). The data represent means ± SEM, *n* = 3 independent experiments. Statistical

significance was determined by using two-way ANOVA with Sidak's multiple comparisons. **e** siRNA transfected U2OS cells were treated with NaAsO₂ (300 μM) for indicated times. The results are representative of three independent experiments. **f** Quantification of immunoblots shown in (**e**). The data represent means ± SEM, *n* = 3 independent experiments. Statistical significance was determined by using two-way ANOVA with Sidak's multiple comparisons. **g** U2OS cells infected with adenovirus expressing TMEM55B-GFP-WT or P66A were incubated with NaAsO₂ (300 μM) alone or NaAsO₂ (300 μM) plus Bafilomycin A1 (200 nM) for indicated times. The results are representative of four independent experiments.
**h** Quantification of immunoblots shown in (**g**). The data represent means ± SEM, *n* = 3 independent experiments. Statistical significance was determined by using two-way ANOVA with Sidak's multiple comparisons. Source data are provided as a Source Data file.

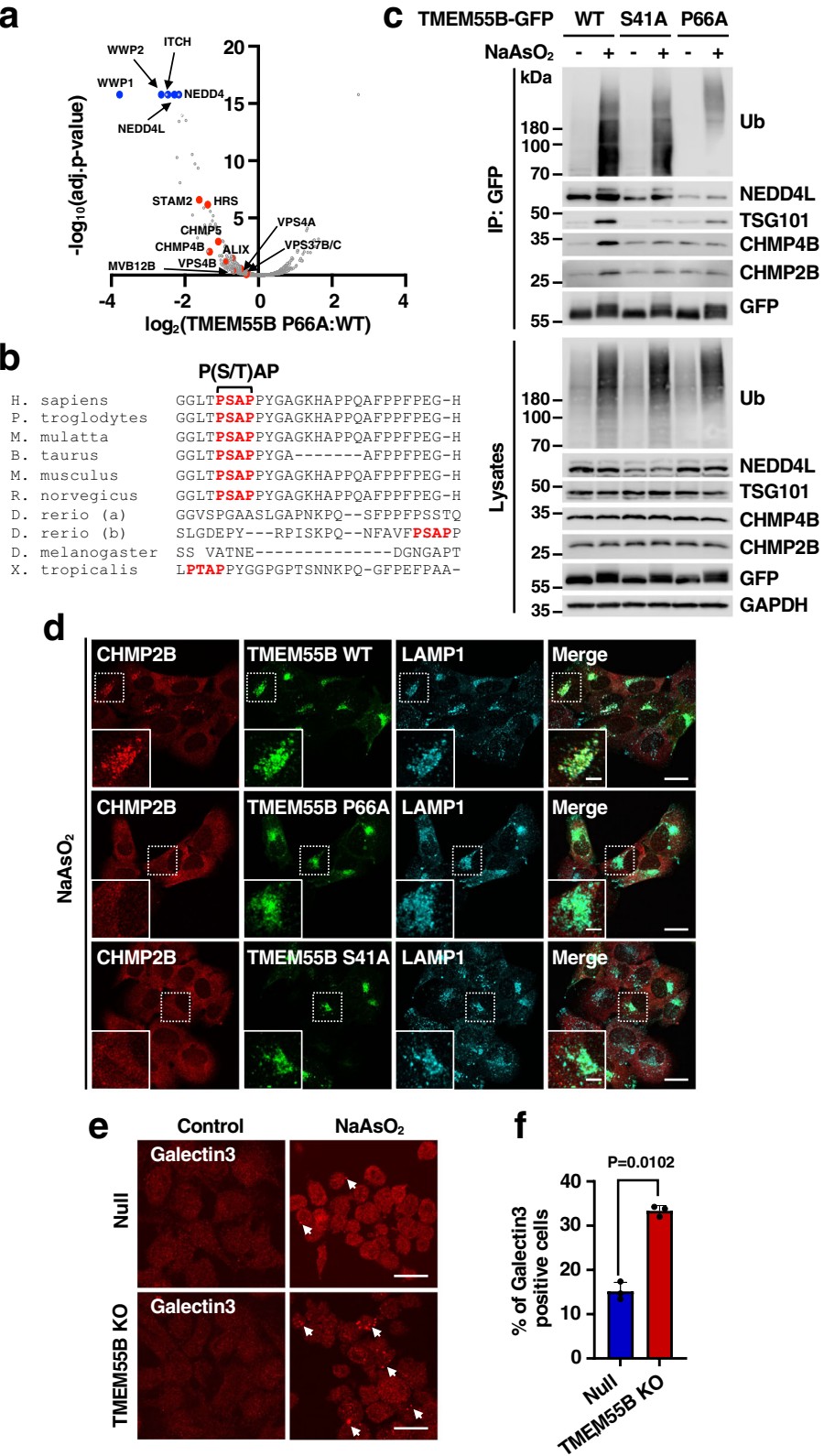

viability under non-stress conditions with minimal cell death detected (Fig. 6a, b). However, after 10 h $NaAsO_2$ treatment, major differences in cell viability were observed, with significantly more necrotic TMEM55B-KO cells compared to controls (Fig. 6a, b). As previously described, incubation with LLOMe at 300 µM for 10 h had minimal effect on cell viability[22] and no differences were observed between Null and TMEM55B-KO cells (Supplementary Fig. 6a, b). These results

indicate that TMEM55B plays an important role in promoting cell viability after oxidative stress.

## TMEM55B sequesters folliculin to induce TFE3 activation

The transcription factors TFEB and TFE3 orchestrate cellular response to a variety of stress conditions. By promoting expression of lysosomal and autophagic genes, TFEB and TFE3 contribute to the elimination of

**Fig. 5 | TMEM55B facilitates lysosomal repair by recruiting the ESCRT complex. a** Volcano plot of hits identified by immunoprecipitation and mass spectrometry analysis of U2OS cells infected with adenovirus expressing TMEM55B-GFP-P66A compared to cells infected with adenovirus expressing TMEM55B-GFP-WT under NaAsO$_2$ (300 μM) treatment. The data were analyzed with two-tailed $t$ test. **b** Multi-sequence alignment of TMEM55B orthologs in different species. The P(S/T)AP motif is marked in red. **c** U2OS cells transfected with plasmid encoding TMEM55B-GFP-WT, S41A or P66A were treated with or without NaAsO$_2$ (300 μM) for 2 h. Cells were lysed and pulled down with GFP beads. The results are representative of three independent experiments. **d** U2OS TMEM55B KO cells transfected with plasmid encoding TMEM55B-GFP WT, S41A, P66A (green) were treated with NaAsO$_2$ (300 μM) for 2 h. Cells were fixed and immunostained with antibodies against CHMP2B (red) and LAMP1 (blue). Scale bars, 20 μm. Inset scale bars, 10 μm. $n$ = 3. **e** Null and TMEM55B KO HeLa cells were treated with or without NaAsO$_2$ (300 μM) for 6 h. Cells were fixed and immunostained with anti-Galectin3 antibody. Scale bars, 20 μm. **f** Quantification of immunofluorescence images shown in (**e**). The data represent means ± SEM. $n$ = 200 cells examined over 3 independent experiments. Statistical significance was determined by using two-tailed unpaired $t$ test. Source data are provided as a Source Data file.

waste material and the restoration of energy homeostasis. We and others have recently reported activation of TFEB and TFE3 in response to oxidative stress both in vivo and in vitro[23–25], as well as their involvement in the expression of antioxidant genes[26]. The Folliculin (FLCN)/Folliculin-interacting protein (FNIP) complex is a critical regulator of TFEB and TFE3 activation. By promoting GDP-loading of RAGC[27], FLCN facilitates interaction of TFEB/TFE3 with RAGs, resulting in retention of the transcription factors in the cytosol[28,29]. Under certain stress conditions, such as treatment with lysosomal ionophores, lipidation of GABARAP serves as a mechanism to sequester the FLCN/FNIP complex, inhibiting its GAP activity and resulting in TFEB and TFE3 activation[30]. Interestingly, we observed increased binding of the FLCN/FNIP complex to both endogenous and recombinant TMEM55B in NaAsO$_2$-treated cells (Fig. 7a, b and Supplementary Fig. 7a, b). The interaction of TMEM55B with FLCN does not requires the PPXY motif (Supplementary Fig. 7a) and is very specific for NaAsO$_2$, as it was not observed in response to other stressors (Fig. 7b and Supplementary Fig. 7b).

These results were confirmed by immunofluorescence analysis. FLCN showed a mainly cytosolic distribution under control conditions (Fig. 7c). Please note that the nuclear staining observed in these cells is nonspecific, as it has been previously detected in FLCN-depleted cells[31,32]. In agreement with previous studies, nutrient deprivation caused clear FLCN recruitment to lysosomes[31–33], which was comparable between Null and TMEM55-KO cells (Fig. 7c, d) Notably, while efficient recruitment of FLCN to lysosomes was observed in Null cells upon treatment with NaAsO$_2$, the amount of FLCN bound to lysosomal membranes was significantly reduced in TMEM55B-depleted cells (Fig. 7c, d). These results indicate that TMEM55B mediates oxidative stress-induced recruitment of FLCN to lysosomes.

We then hypothesized that sequestration of FLCN by TMEM55B may constitute a mechanism to promote TFEB and TFE3 activation. To test this, we analyzed TFE3 activation in Null and TMEM55B-KO cells by immunofluorescence and immunoblot. As seen in Supplementary Fig. 7, translocation of TFE3 from the cytosol to the nucleus in response to NaAsO$_2$ was significantly less efficient in TMEM55B-KO cells (Supplementary Fig. 7c, d). In contrast TFE3 activation by incubation with the mTOR inhibitor, Torin-1, was not affected by depletion of TMEM55B (Supplementary Fig. 7e, f). Accordingly, dephosphorylation of TFE3-S321, which is required for the dissociation of the TFE3/14-3-3 complex and consequent TFE3 nuclear translocation[31], was inhibited in TMEM55B-KO cells in response to NaAsO$_2$ but not Torin-1 (Fig. 7d, e). Furthermore, mTOR activity was not affected by depletion of TMEM55B (Fig. 7d). Altogether, our results suggest that TMEM55B modulates non-canonical activation of TFEB and TFE3 in response to oxidative stress.

### Increased susceptibility of *tmem55*-KO embryos to arsenite
Next, we sought to corroborate the role of TMEM55B in response to oxidative stress in vivo. Since depletion of TMEM55B in mice was shown to be embryonic lethal[34], we decided to use zebrafish as model system. In zebrafish, two paralogs of the mammalian *TMEM55B* gene are present, indicated as *tmem55ba* and *tmem55bb* (Fig. 8). We simultaneously targeted the exon 2 of both genes using a multiplex CRISPR/Cas9 genome editing approach[35], and were able to obtain

double-heterozygous F1 progeny. The initial genotyping analysis by fluorescent PCR followed by Sanger sequencing confirmed the presence of frame-shift mutations at the genomic level, two deletions of 13 and 16 bases in *tmem55ba* and *tmem55bb* genes, respectively (Fig. 8a). Notwithstanding the presence of potentially deleterious mutations at the genomic level, double homozygous fish, hereafter indicated as *tmem55b*-KO, were viable and fertile and did not present any evident phenotype. As expected, qPCR analysis showed that *tmem55b*-KO embryos presented a strong reduction of both *tmem55b* transcripts at 5 days post fertilization (dpf) (Supplementary Fig. 8a). To prevent potential compensation from the family member *tmem55a*, we injected specific gRNAs targeting *tmem55a* exons 1 and 4 directly in double homozygous *tmem55b*-KO embryos (Fig. 8a). Again, through genotyping followed by Sanger sequencing, we confirmed the presence of frame-shift mutations on the same allele (two deletions of 2 and 5 bases in exon 1 and exon 4, respectively) in the *tmem55a* gene and the mRNA transcripts (Fig. 8a). As previously observed for the *tmem55b*-KO fish, the genetically introduced mutations induced the activation of nonsense-mediated mRNA decay (NMD) mechanisms and, therefore, a reduction of the mRNA transcript levels (Supplementary Fig. 8b). Triple homozygous *tmem55* embryos (*tmem55*-KO) grew normally without noticeable embryonic defects (Fig. 8b), reached adulthood, and were fertile. To test the possibility that eventual phenotypes appear only in response to cellular stress[36], we treated WT and *tmem55*-KO embryos continuously with 2 mM NaAsO$_2$ and monitored their survival rate during the treatment. As shown in Fig. 8c, *tmem55*-KO embryos were significantly more susceptible to prolonged NaAsO$_2$ treatment and presented a lower survival rate compared to controls, confirming the important involvement of *tmem55* in cellular stress response to increased levels of oxidative stress.

## Discussion
The role of TMEM55B in regulating lysosomal positioning is well established. Through its interaction with JIP4, TMEM55B connects lysosomes to the minus-end microtubule motor dynein, allowing retrograde movement of lysosomes from the periphery to the cell center. This function is particularly relevant during starvation conditions, when lysosomal retrograde movement facilitates contact and fusion of lysosomes with autophagosomes. Yet, in recent years, the identification of an increasing repertoire of protein complexes implicated in the dynamic regulation of lysosomal positioning[37] has suggested some level of specialization, with certain complexes playing a more relevant role in specific cell types, lysosomal populations, or in response to different types of stress.

Here we show that TMEM55B function is stress dependent. While the interaction between TMEM55B and JIP4 is critical to promote lysosomal retrograde transport under nutrient deprivation, the complex dissociates in response to NaAsO$_2$. Paradoxically, treatment with NaAsO$_2$ causes robust JIP4-dependent clustering of lysosomes at the cell center[7]. This may be explained by recent observations showing that JIP4 phosphorylation following oxidative stress induces its interaction with the TRPML1/ALG2 complex, which also mediates dynein-dependent retrograde transport of lysosomes[9,38,39]. Furthermore, JIP4 was also showed to link ARL8, RUFY3 and RUFY4 to dynein to regulate

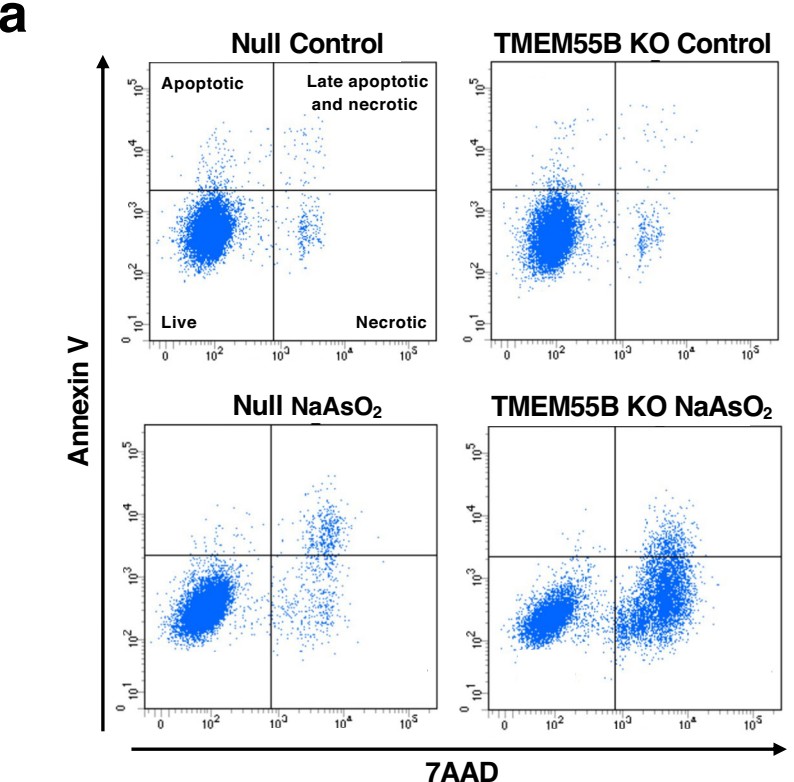

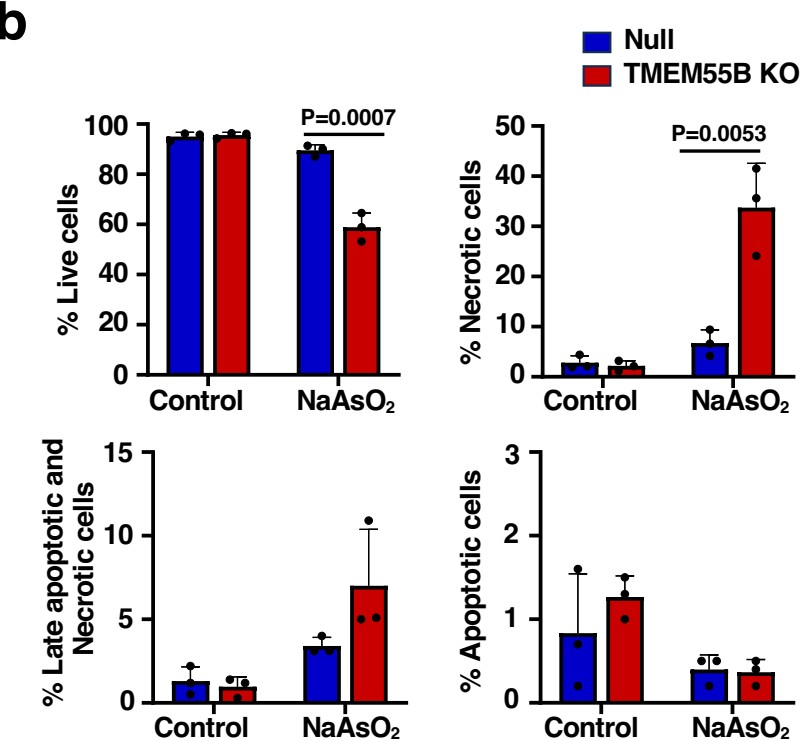

**Fig. 6 | Depletion of TMEM55B reduces cell viability under oxidative stress conditions. a** Null and TMEM55B KO U2OS cells were treated with or without NaAsO₂ (300 μM) for 10 h. Cells were analyzed by Flow cytometry with Annexin V and 7AAD. 7AAD+ are necrotic, Annexin V+ are apoptotic and Annexin V + /7AAD+ are late apoptotic and necrotic cells. **b** Quantification of the population of live, necrotic (7AAD + ), apoptotic cells (Annexin V + ) and late apoptotic and necrotic cells (Annexin + /7AAD + ) from (**a**). The data represent means ± SEM, $n = 3$ independent experiments. Data taken from three independent experiments and statistical significance was determined by using two-way ANOVA with Sidak's multiple comparisons. Source data are provided as a Source Data file.

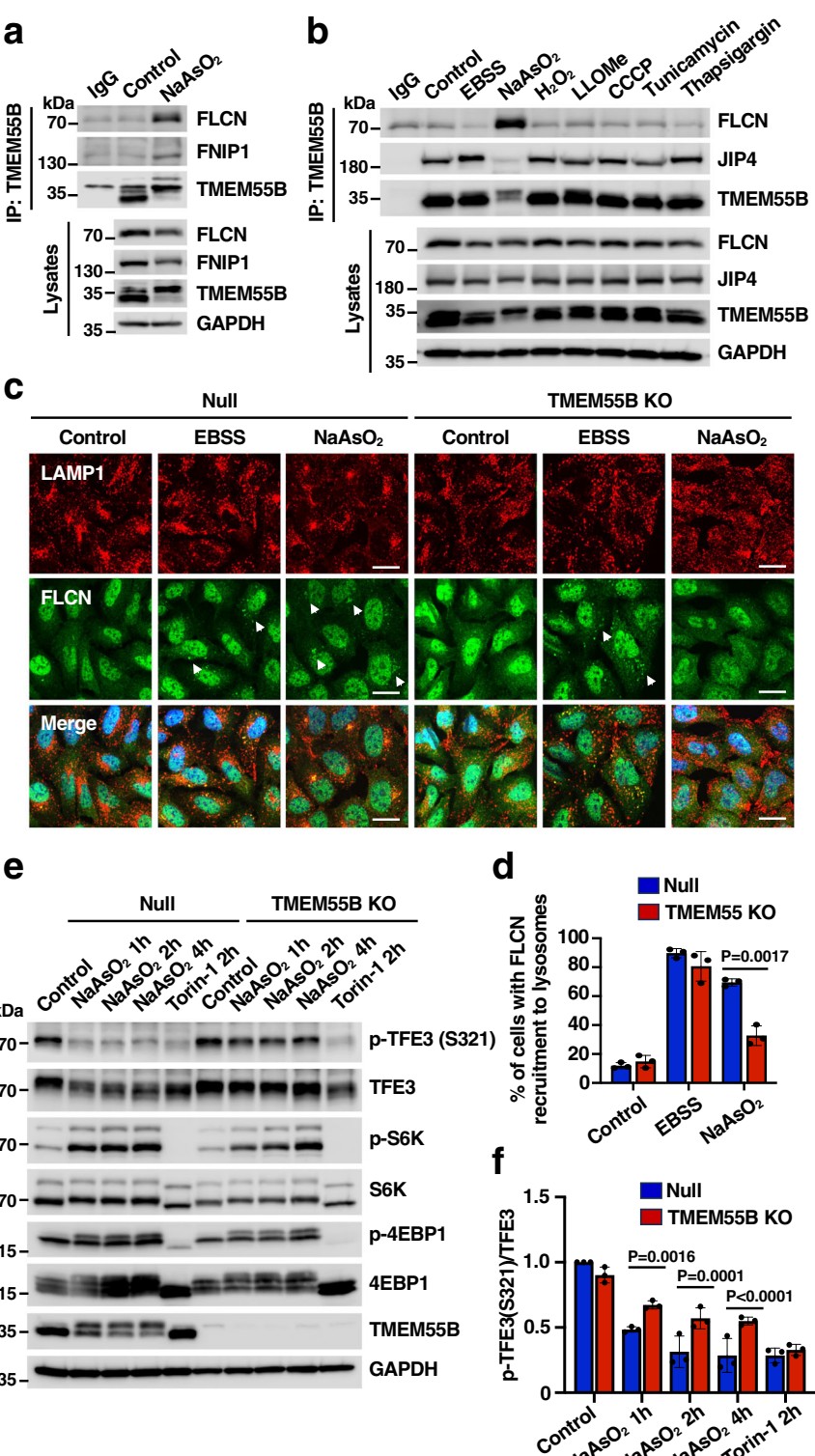

**Fig. 7 | TMEM55B promotes TFE3 activation through FLCN sequestration.**
**a** U2OS cells treated with or without NaAsO₂ (300 µM) for 2 h were lysed and immunoprecipitated with anti-TMEM55B antibody. The results are representative of three independent experiments. **b** U2OS cells treated with various drugs were lysed and immunoprecipitated with anti-TMEM55B antibody. EBSS for 4 h, NaAsO₂ (300 µM) for 2 h, H₂O₂ (500 µM) for 4 h, LLOMe (1 mM) for 2 h, CCCP (25 µM) for 4 h, Tunicamycin (10 µg/ml) for 4 h, Thapsigargin (10 µM) for 4 h. The results are representative of three independent experiments. **c** Null and TMEM55B KO U2OS cells were treated with either EBSS for 4 h or NaAsO₂ (300 µM) for 2 h. Cells were fixed and immunostained with antibodies against LAMP1 (red) and FLCN (green).

Scale bars, 20 µm. n = 3. **d** Quantification of immunofluorescence images shown in (**c**). The data represent means ± SEM, n = 200 cells examined over 3 independent experiments. **e** Null and TMEM55B KO U2OS cells were treated with either NaAsO₂ (300 µM) or Torin-1 (250 nM) for the indicated times. Samples were analyzed by immunoblot with the indicated antibodies. The results are representative of three independent experiments. **f** Quantification of immunoblots shown in (**e**). The data represent means ± SEM, n = 3 independent experiments. Statistical significance was determined by using two-way ANOVA with Sidak's multiple comparisons. Source data are provided as a Source Data file.

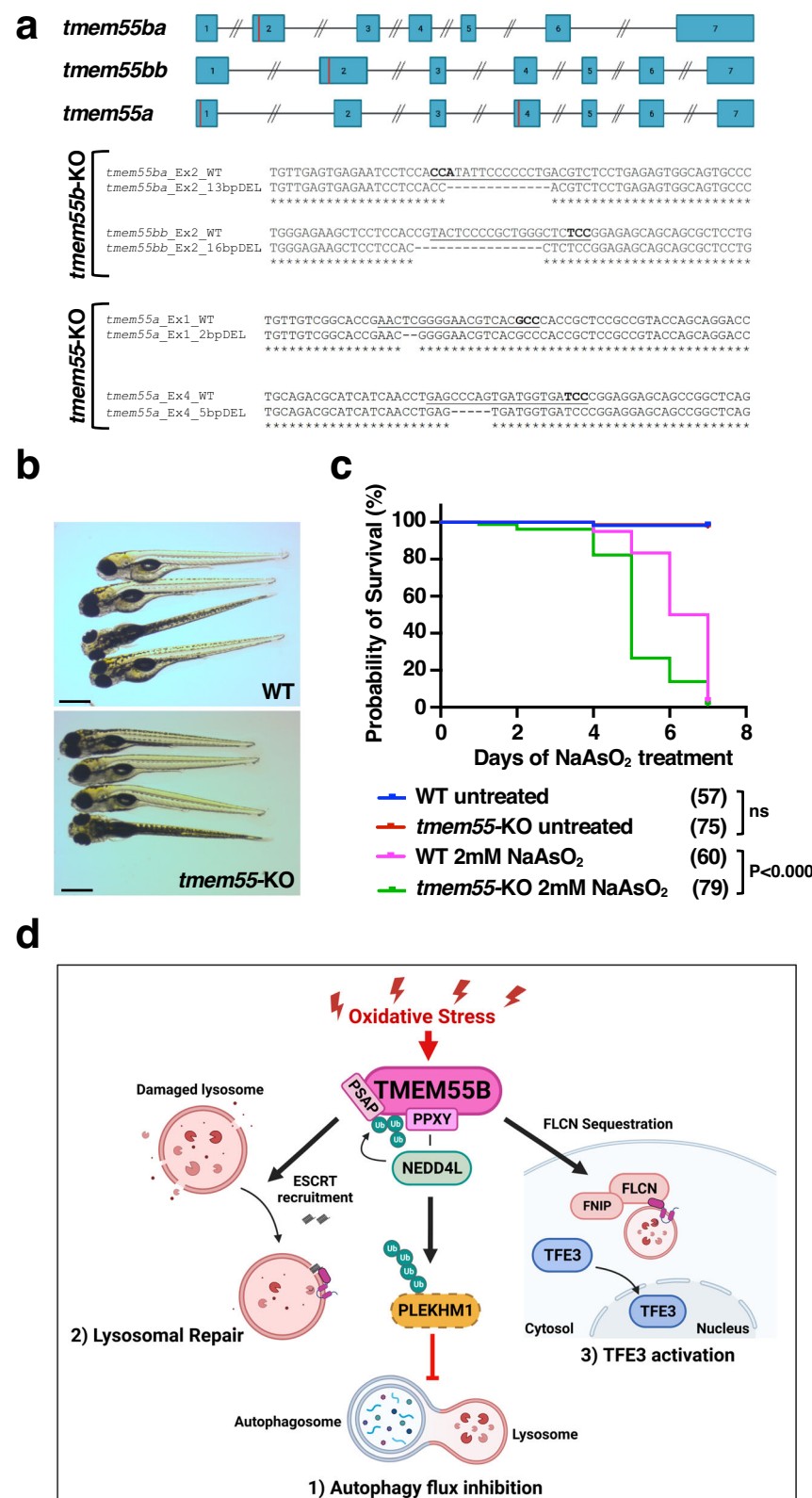

lysosomal size and positioning[40,41]. The ability of JIP4 to select particular lysosomal cargo depending on stress is quite intriguing and could reflect functional differences (retrograde transport versus docking) or JIP4-mediated modulation of specific lysosomal signaling complexes.

Under oxidative stress, TMEM55B coordinates cellular responses that contribute to restore lysosomal integrity and cellular homeostasis (Fig. 8d). TMEM55B interacts with members of the NEDD4-like family of E3 ligases via a PPXY motif, inducing their recruitment to the lysosomal surface. The observed NaAsO$_2$-induced NEDD4L activation may further facilitate efficient ubiquitination of target proteins. TMEM55B itself is a target of NEDD4-like E3 ligases. This ubiquitination, together with the presence of a PSAP motif, promotes the recruitment of ESCRT proteins and facilitates lysosomal repair. As such, NaAsO$_2$-induced

**Fig. 8 | Generation and characterization of *tmem55*-KO mutants. a** (Top panel) Schematic representation of zebrafish *tmem55ba*, *tmem55bb* and *tmem55a* genes with exons shown as blue rectangles connected by introns. Red bars show the target region of the sgRNAs used for CRISPR/Cas9 knockout. Neither exons nor introns are drawn to scale. (Bottom panel) Nucleotide sequences of CRISPR target regions in exons 2 of *tmem55ba* and *tmem55bb* and exons 1 and 4 of *tmem55a* genes. sgRNAs are underlined, PAM sites are in bold and deleted nucleotides in the mutant alleles are marked by dashes. **b** Representative pictures of WT and *tmem55*-KO at 5

dpf. All embryos are depicted with anterior to the left. Scale bars = 500 μm. **c** Kaplan-Meier curves showing the survival rates in zebrafish embryos following continuous exposure to 2 mM $NaAsO_2$ from 1 dpf stage. Untreated embryos are used as controls. The number of embryos per group is indicated in parenthesis. (Logrank test, $P < 0.0001$). Data in (**b**) and (**c**) show one representative experiment out of three independently performed. **d** Model depicting the proposed role of TMEM55B orchestrating cellular responses to oxidative stress. Illustration created with BioRender.com. Source data are provided as a Source Data file.

accumulation of galectin3-positive puncta, a marker for rupture of lysosomal membranes, as well as necrosis, were significantly increased in TMEM55B-depleted cells. In contrast, galectin3 staining in response to LLOMe was not affected by depletion of TMEM55B, suggesting that cells may use distinct machineries and regulatory mechanisms for different damaging agents. However, some commonalities remain. For example, translocation of ESCRT to lysosomes in response to LLOMe is triggered by the efflux of lysosomal $Ca^{2+}$ and subsequent recruitment of ALIX, which in turn brings other ESCRT-III components[42]. The ESCRT-I subunit TSG101 is also required for efficient ESCRT-III recruitment[22]. Likewise, TMEM55B recruits TSG101, presumably via its P(S/T)AP motif, while its NEDD4-dependent ubiquitination may further facilitate recruitment of the ESCRT machinery.

Unexpectedly, we observed TMEM55B- and NEDD4-dependent ubiquitination of PLEKHM1, causing PLEKHM1 proteasomal degradation and reducing the rate of fusion between autophagosomes and lysosomes. These results may seem contra intuitive, as autophagy induction is important to eliminate protein aggregates and damaged organelles generated by oxidative stress, and to maintain cell viability. However, fusion of autophagosomes with unhealthy lysosomes may be contra productive, as the constant delivery of cargo to overloaded or non-functional lysosomes will likely result in accumulation of undegraded material. Therefore, slowing down the rate of autophagosome degradation, while at the same time promoting lysosomal repair and activation of stress pathways, may be beneficial for the cell. Furthermore, by specifically targeting the PLEKHM1 associated with lysosomal membranes, cells may preferentially prevent fusion of autophagosomes only with those lysosomes that are damaged. It is tempting to speculate that NEDD4-like E3 ligases might ubiquitinate additional lysosomal transmembrane proteins that misfold because of the stress, thus promoting their ESCRT-dependent removal as part of a protein quality control mechanism.

TMEM55B also activates transcriptional programs in response to lysosomal perturbations. TMEM55B-dependent sequestration of the FLCN/FNIP complex following $NaAsO_2$ treatment results in translocation of TFE3 to the nucleus. In the nucleus, TFE3 can induce expression of multiple antioxidant[26] and lysosomal genes[31], contributing to cell survival and allowing restoration of cellular homeostasis. These data are consistent with recent studies showing that inhibition of the FLCN/FNIP complex is sufficient to induce TFEB and TFE3 activation, even in conditions in which mTORC1 remains active. For example, under metabolic stress, AMPK-dependent phosphorylation of FNIP1 results in inhibition of the FLCN/FNIP complex GAP activity, leading to accumulation of GTP-loaded RagC and consequent TFEB and TFE3 nuclear translocation[43]. Likewise, sequestration of FLCN/FNIP in GABARAP-conjugates membrane compartments prevents mTORC1-dependent inactivation of TFEB[30]. Our results further demonstrate that uncoupling FLCN/FNIP-mediated RagC/D regulation may be a more general mechanism of TFEB/TFE3 activation than previously anticipated. One additional and important consideration is that only those agents capable of generating high ROS levels, such as $NaAsO_2$ and acrolein, caused TMEM55B phosphorylation and JIP4 dissociation. This suggests that activation of specific kinases may be critical to link lysosomal damage to TMEM55B-mediated stress responses.

Finally, the critical contribution of TMEM55B to oxidative stress response was corroborated in vivo. The viability of *tmem55*-knockout

zebrafish embryos incubated with $NaAsO_2$ was severely reduced when compared with WT embryos, indicating that TMEM55B has a protective role in arsenic-mediated embryo toxicity. In summary, our data identify TMEM55B as a molecular sensor that coordinates many fundamental lysosomal processes, including autophagosome degradation, lysosomal repair, and activation of stress responses.

## Methods

### Ethical statement
All zebrafish experiments were performed in compliance with the National Institutes of Health (NIH) guidelines for animal handling and research under NHLBI Animal Care and Use Committee (ACUC) approved protocol H-0252R5.

### Zebrafish husbandry and imaging
Zebrafish husbandry and embryo staging were performed in accordance with relevant guidelines and regulations. During development the living embryos were observed with a Zeiss Discovery.V12 stereomicroscope (Zeiss) with a PlanApo S 0.63X lens (Zeiss). Pictures were taken with an AxioCam ERc5s in Zen 2.3 Pro software (Zeiss).

### Generation of zebrafish TMEM55 KO lines
TMEM55B knockout (KO) mutants were generated using CRISPR/Cas9 methods[35,44]. Briefly, specific gRNAs were co-injected with Cas9 mRNA in the yolk of TAB5 embryos at the 1-cell stage. At 24 h post fertilization (hpf) -32 embryos from each injection were collected, the genomic DNA was extracted using the Extract-N-Amp Tissue PCR kit (Sigma-Aldrich, XNAT2) and the target efficacy was tested using Fluorescent-PCR methods[44]. Only embryos from injections with high target frequency were moved into the system and grown until adulthood. The $F_1$ generations obtained from an outcross of potential $F_0$ founders with WT fish were screened for mutations and raised to adulthood. $F_1$ adults were genotyped by fin amputation, the gDNA was extracted by the fin tissues and fluorescent PCRs were performed[35]. To determine the exact mutations in the $F_1$ population the fluorescent PCR reactions were Sanger sequenced and the exact sequence was reconstructed by manually annotation of the electropherograms using WT sequences as reference. Double-heterozygous TMEM55B-KO fish were in-crossed and double-homozygous animals were selected by genotyping. Finally, TMEM55B-KO were incrossed and the embryos were injected with specific gRNAs targeting the TMEM55A gene to obtain the triple-KO animals. For a list of the targets used see Supplementary Table 1.

### Cell lines cultures and treatments
U2OS cells (ATCC, HTB-96) and HeLa cells (ATCC, CCL-2) were grown at 37 °C, 5% $CO_2$ in DMEM, high glucose, GlutaMAX, sodium pyruvate (Gibco, 10569044) supplemented with 10% fetal bovine serum (Invitrogen, 21041–025), 100 U/ml penicillin and 100 μg/ml streptomycin (Gibco, 2114).

For TMEM55B KO stable cell lines, U2OS and Hela cells were generated with the Edit-r-lentiviral system (Horizon Discovery) with gRNA sequences targeted against TMEM55B or negative control non-target sequences according to the manufacturers protocol. In brief, cells grown on a 24 well plate were transduced with hEF1a promoter containing Cas9 lentiviral particles (Horizon Discovery, VCAS10126) at MOI = 0.3 for 6 h in DMEM to generate stable Cas9 containing cells.

After 24 h, cells containing the Cas9 nuclease were selected with blasticidin for 5 days. Next, Cas9 stable cells were transduced with gRNA lentiviral particles targeted against TMEM55B (Horizon Discovery, GSGH11838-246567343) or NT gRNA (Horizon Discovery, GSG11811) at MOI = 0.3 for 6 h in DMEM. 24 h after incubation gRNA lentiviral containing cells were selected with puromycin for 5 days. TMEM55B or NT gRNA containing cells were then plated into single cell colonies on a 96 well plate and grown until confluency. TMEM55B KO clones were verified by sequencing and western blotting analysis.

For drug treatment experiments, cells were incubated for the indicated periods of time at 37 °C in medium containing the following reagents: DMSO (Invitrogen, D12345), Torin-1 (TOCRIS, 4247/10), Sodium arsenite solution (Sigma-Aldrich, 1062771000), ERK inhibitor: U0126 (Selleckchem, S1102), JNK inhibitor: JNK inhibitor VIII (Selleckchem, S7794), p38 inhibitor: SB203580 (Selleckchem, S1076), JAK3 VI inhibitor (Sigama-Aldrich, 420126), MG-132 (Sigma-Aldrich, M7449), Bafilomycin A1 (Cells signaling Technology, 54645), Hydrogen peroxide solution 30% (w/w) (Sigma-Aldrich, H1009), LLOMe (Cayman, 6491-83-4), CCCP (Cayman, 25458), Tunicamycin (Cell signaling Technology, 12819 S), Thapsigargin (Cell signaling Technology, 12758 S), Acrolein (Restek, 30645), Spermidine (Sigma-Aldrich, S0266), TAK243 (Selleckchem, S8341). For starvation experiments, cells were washed three times in PBS and incubated for 4 h at 37 °C in Earle's balanced salt solution (Gibco, 24010043). For washout experiments, cells were washed three times in PBS and incubated for indicated times at 37 °C in medium.

## Antibodies

The following antibodies were used in this study: anti-TFE3 (Sigma, HPA023881), anti-phospho S321 TFE3 (YenZym Antibodies), anti-Flag (clone M2, Sigma-Aldrich, F1804), anti-LAMP1 from the Developmental Studies Hybridoma Bank deposited by August, J.T. (DSHB, 1D4B), anti-TMEM55B (Proteintech, 23992-1-AP), anti-CHMP2B (Proteintech, 12527-1-AP), anti-CHMP4B (Proteintech, 13683-1-AP), anti-TSG101 (Proteintech, 14497-1-AP), anti-GAPDH (Santa Cruz Biotechnology, sc-365062), anti-Ub (Santa Cruz Biotechnology, sc-8017), anti-VPS41 (Santa Cruz biotechnology, sc-377271), anti-Galectin3 (Santa Cruz Biotechnology, sc-32790), anti-phospho-ITCH (Sigma-Aldrich, AB10050), anti-ITCH (Cell signaling Technology, 12117), anti-PLEKHM1 (Cell signaling Technology, 77092), anti-NEDD4 (Cell signaling Technology, 2740), anti-NEDD4L (Cell signaling Technology, 4013), anti-JIP4 (Cell signaling Technology, 5519), anti-LC3B (Cell signaling Technology, 43566), anti-phospho-p70 S6 Kinase (Cell Signaling Technology, 9205), anti-p70 S6 Kinase (Cell Signaling Technology, 2708), anti-phospho-4E-BP1 (Cell Signaling Technology, 2855), anti-4E-BP1 (Cell Signaling Technology, 9644), anti-phospho-p38 (Cell Signaling Technology, 4511), anti-phospho-c-Jun (Cell signaling Technology, 3270), anti-phospho-ERK (Cell signaling Technology, 9101), anti-LAMTOR1 (Cell Signaling Technology, 8975), anti-FNIP1 (Cell signaling Technology, 36892), anti-FLCN (Cell Signaling Technology, 3697), HRP-conjugated anti-rabbit IgG (Cell Signaling Technology, 7074), HRP-conjugated anti-mouse (Cell Signaling Technology, 7076), Alexa Fluor 568-conjugated goat anti-mouse IgG (Invitrogen, A21090), Alexa Fluor 488-conjugated goat anti-rabbit IgG (Invitrogen, A-11008) Alexa Fluor 488-conjugated goat anti-mouse IgG (Invitrogen, A-11001). For immunoblotting, antibodies were diluted as 1:1000. For immunofluorescence, antibodies were diluted from 1:50 to 1:1000.

## Recombinant DNA plasmids and adenovirus

TMEM55B-GFP expression vector was generated by cloning the full-length encoding sequence of human TMEM55B obtained by PCR amplification followed by in-frame cloning into EcoRI-SalI sites of pmEGFP-C2[7]. The TMEM55B CD construct was generated by the insertion of a stop codon after the tyrosine 207 codon of TMEM55B-GFP expression vector. Amino acid substitutions in TMEM55B were made using the QuickChange Lightning site-directed mutagenesis kit (Stratagene) according to the manufacturer's instructions[7]. PLEKHM1-Flag expression vector (Addgene, 73593) was used to generate the truncated mutants by PCR amplification. RAB7A-HA expression vector was obtained from Addgene (Addgene, 131417) and RAB7-GFP expression vector was generated by cloning into HindIII-KpnI sites of pmPEGF-C1. Adenovirus expressing TMEM55B-WT-GFP and TMEM55B-P66A-GFP were prepared, amplified, and purified by Welgen, Inc.

## DNA transfection

Plasmid transfections were performed using Lipofectamine 2000 (Invitrogen, 11668019) or Fugene 6 (Roche, E2691), according to manufacturer's instructions.

## RNA interference (RNAi)

For siRNA depletion, cells were transfected using Lipofectamine RNAiMAX transfection reagent (Invitrogen, 13778075) with ON-TARGETplus non-targeting pool siRNA duplexes or ON-TARGETplus smart pool siRNA duplexes targeted against TMEM55A and TMEM55B genes (Dharmacon-Thermo Scientific, L-013808-01-0005 and L-016425-02-0005 respectively). Transfected cells were analyzed 72 h after transfection.

## Immunoprecipitation and immunoblotting

Cells were washed with cold PBS and lysed in lysis buffer containing 150 mM NaCl, 20 mM HEPES, 5 mM EDTA, 10% Glycerol, 1% Triton X-100 with protease inhibitor (Roche, 11836170001) and phosphatase inhibitor (Roche, 4906837001). Whole cell lysates were homogenized and incubated on ice for 30 min and then centrifuged at 16,000 x g for 15 min at 4 °C. For immunoprecipitation, the soluble fractions were incubated with 1 μg of antibody for overnight at 4 °C. The Antibody-protein complexes were incubated with protein G-Sepharose beads (GE Healthcare, GE17-0618-01) for 2 h at 4 °C and collected, washed three times with lysis buffer and then proteins were mixed with NuPage 4X loading buffer (Life Technologies, NP0007).

For Flag or GFP pull down, whole cell lysates were collected as indicated above and soluble fractions were incubated with M2 anti-Flag magnetic beads (Sigama-Aldrich, M8823) or GFP-Trap agarose (ChromoTeck, gta) for 1 h at 4 °C and collected, washed three times with lysis buffer and then proteins were mixed with NuPage 4X loading buffer. Samples were analyzed by SDS-PAGE 4–20% gradient gels (Invitrogen, XP04202BOX) and transferred to nitrocellulose membranes. Membranes were immunoblotted with the indicated antibodies. HRP-conjugated secondary antibodies were used at a dilution of 1:5000. Immunoblots were developed with Radiance Plus Chemiluminescent Substrate (Azure Biosystems, 10147-298) and exposed using a GE Healthcare Life Sciences Amersham Imager 600. Immunoblots were quantitated with densitometric analysis using ImageJ (NIH) and normalized to GAPDH as a loading control.

## Immunofluorescence confocal microscopy

U2OS cells grown on glass coverslips were washed three times with PBS and fixed with 4% formaldehyde (Electron Microscopy Sciences, 15710) diluted in PBS for 15 min at room temperature. After fixation, cells were washed with PBS and incubated with 1% BSA for 10 min. Cells were then incubated with primary antibodies for 1 h at room temperature in IF buffer PBS containing 1% BSA and 0.1% (w:v) saponin (Sigma-Aldrich, S-4521). Cells were washed three times with PBS and incubated with secondary antibodies for 30 min at room temperature followed by an additional three times wash in PBS. Coverslips were mounted with ProLong Diamond Antifade Mountant reagent with DAPI (Invitrogen, P36966). Images were acquired with an LSM 510 Meta confocal microscope (Zeiss, Oberkochen, Germany) with 63x numerical aperture 1.4 oil immersion objective with a Zeiss AxioCam camera. For Galectin3 staining, Hela cells grown on glass coverslips

were washed two times with PBS and fixed with cold methanol for 10 min at −20 °C. Cells were then incubated with Galectin3 antibody in IF buffer and followed by immunostaining as indicated above. Colocalization between ubiquitin and TMEM55B was measured using Pearson's correlation coefficient calculator tool with the Imaris x64 9.9.0.

## Mass spectrometry

U2OS cells infected with adenovirus expressing TMEM55B-WT-GFP or TMEM55B-P66A-GFP cells treated with or without 300 μM $NaAsO_2$ for 2 h. Cell lysates were subjected to immunoprecipitation by using GFP beads. Immunoprecipitated proteins were subjected to SDS–PAGE, and gel bands from corresponding molecular weights were excised for enzymatic digestion. Briefly, the samples were first reduced with TCEP (tris(2-carboxyethyl)phosphine, Sigma-Aldrich) and alkylated with CAA (chloroacetamide, Sigma-Aldrich), and then digested with chymotrypsin (Promega). The resulting peptide mixtures were analyzed with an Orbitrap Fusion Lumos that is equipped with a Dionex Ultimate 3000 nanoLC system (ThermoFisher Scientific). Peptide IDs and phosphorylation sites were assigned with Mascot V2.5 (Matrix Science). The confidence of phosphorylation site localization is assessed with ptmRS node in Proteome Discoverer 2.2 platform (ThermoFisher Scientific). All peptides were filtered out at 1% false discovery rate (FDR) and their relative abundances were compared based on the areas under curve (AUC) of their corresponding chromatographic peaks.

## Flow cytometry analysis

For Annexin V apoptosis Detection, U2OS cells were washed with PBS and treated with TrypLE express enzyme (Gibco, 12605010) for less than 3 min at 37 °C. Subsequently, the cells were incubated with eBioscience Annexin V Apoptosis Detection Kit (Invitrogen, 88-8006-72) according to the manufacturer's instructions. Cells were then analyzed by flow cytometry using a BD Fortessa cytometer. For Intracellular ROS analysis, U2OS cells were washed, trypsinized and collected. Cells were diluted with $H_2DCFDA$ (20 μM; Invitrogen, C400) for 30 min and then incubated with $NaAsO_2$ (300 μM) for 2 h, $H_2O_2$ (500 μM) for 4 h, Acrolein (200 μM) for 2 h or Spermidine (300 μM) for 4 h. After incubation, cells were washed with PBS and flow cytometry was performed by using BD LSR Fortessa. A figure exemplifying the gating strategy is provided in the Supplementary Information (Supplementary Fig. 9).

## Lambda phosphatase assay

U2OS cells were treated with or without $NaAsO_2$ (300 μM) for 2 h. Cells were washed with cold PBS and lysed in lysis buffer containing 150 mM NaCl, 20 mM HEPES, 5 mM EDTA, 10% Glycerol, 1% Triton X-100 with protease inhibitor (Roche, 11836170001). Whole cell lysates were homogenized and incubated on ice for 30 min and then centrifuged at 16,000 x $g$ for 15 min at 4 °C. 1 μl of Lambda Protein phosphatase (NEB, P0753S), 10 mM $MnCl_2$ and 10x NEB Buffer was added to lysate and incubated 30 min at 30 °C. For immunoprecipitation, whole cell lysates were immunoprecipitated with anti-TMEM55B for overnight and incubated with protein G beads for 2 h. After washing beads, 1 μl of Lambda Protein phosphatase, 10 mM $MnCl_2$ and 10x NEB Buffer was added to beads and incubated 30 min at 30 °C.

## Ubiquitination assay under 2% SDS denaturing condition

Cells were washed with TBS and treated with TrypLE express enzyme (Gibco, 12605010). Cells were collected with TBS and centrifuged at 500 g for 5 min at 4 °C. Cell pellets were dispersed by pipetting with TBS containing 2% SDS and immediately boiled at 95 °C for 10 min. Then lysis buffer containing 150 mM NaCl, 20 mM HEPES, 5 mM EDTA, 10% Glycerol, 1% Triton X-100 with 10 mM deubiquitinases inhibitor NEM (Thermo Scientific, 23030), protease inhibitor (Roche,

11836170001) and phosphatase inhibitor (Roche, 4906837001) was added. Whole cell lysates were incubated on ice for 10 min, centrifuged at 16,000 x $g$ for 15 min at 4 °C and subsequently immunoprecipitated as described above.

## Total RNA extraction from zebrafish embryos and cDNA synthesis

Fifty zebrafish embryos at each stage were euthanized and sacrificed. Embryos were grossly homogenized using a 1 ml syringe with a 25 G needle followed by further homogenization using the QIAshredder homogenizers (Qiagen). Total RNA was extracted using the PureLink RNA Mini Kit (Invitrogen) and digested with DNase I and the cDNA synthesis was carried out with the Superscript III First-strand Synthesis SuperMix for qRT-PCR (Invitrogen) using 1 μg of total RNA from each sample following standard conditions. All the primer pairs were designed on different exons to avoid the amplification of DNA possibly contaminating cDNA preparations. RT-PCR products were then separated on agarose gels at various concentrations (from 1 to 3% maximum, based on the fragments length) and visualized by ethidium bromide staining. Lack of gDNA contamination in cDNA samples was confirmed using a fragment of zebrafish β-actin cDNA that was amplified by PCR (35 cycles) as previously described[45].

## Quantitative real-time PCR

Quantitative real-time PCR analyses were performed in quintuplicate using SYBR Green PCR Master Mix (Applied Biosystems) and primers designed using Primer3web (version 4.1.0). Reactions were assembled in 384-well plates (Applied Biosystems) and run under standard conditions on a QuantStudio™ 12 K Flex Real-Time PCR System (Applied Biosystems). Each experiment was replicated at least three times. Specificity of the RT-PCR products was assessed by gel electrophoresis. A single product with the expected length was detected for each reaction.

The 2 − ΔΔCt method with an RQmin/RQmax confidence set at 99% was applied. The error bars indicate the calculated relative quantity maximum (RQmax) and minimum (RQmin) expression levels that represent the standard errors of the mean expression level (RQ value). The upper and lower limits define the region of expression within which the true expression level value is likely to occur. See Supplementary Table 1 for the primer sequences.

## $NaAsO_2$ treatment and survival assay in zebrafish embryos

WT and TMEM55-KO fish were crossed and embryos at 24 hpf were moved to E3 medium and divided in different 60 mm tissue culture dishes (Falcon) based on the size of each clutch (usually ~70 embryos/60 mm dish or ~100 embryos/100 mm dish). The medium was removed and changed with a 2 mM solution of $NaAsO_2$ (ChemCruz, CAS number 7784-46-5) in E3 medium from a 1 mM stock solution in ultra-pure water. Every morning the survival of the embryos was checked under a stereomicroscope (Zeiss), the dead embryos were removed, and the solutions were changed with new ones prepared fresh every day from a 1 M $NaAsO_2$ stock solution prepared daily diluting the powder in ultra-pure water. At the end of each treatment the survival embryos were anesthetized and euthanized. The collected data were used to calculate the survival rate using the Kaplan-Meier method in Prism 9 (Graphpad). Statistical analysis was performed on pairwise comparison of individual survival curves using both the log-rank (Mantel-Cox) test and the Gehan-Breslow-Wilcoxon test, obtaining the same results.

## Statistics and reproducibility

Each experiment was performed at least three times independently with similar results unless otherwise mentioned in figure legends. Obtained data were processed in Excel (Microsoft Corporation) and Prism 9 (GraphPad Software) to generate bar charts and perform

statistical analyses. Student's *t* test, one-way ANOVA or two-way ANOVA were run for each dependent variable, as specified in each figure legend. All data are presented as mean ± SEM.

No statistical method was used to predetermine the sample size. No data were excluded from the analyses. The experiments were not randomized. The Investigators were not blinded to allocation during experiments and outcome assessment.

### Reporting summary
Further information on research design is available in the Nature Portfolio Reporting Summary linked to this article.

## Data availability
The mass spectrometry proteomics data generated in this study have been deposited to the ProteomeXchange Consortium via the PRIDE[46] partner repository with the dataset identifier PXD043609 (http://www.ebi.ac.uk/pride/archive/projects/PXD043609).

The data set identifier for the identification of TMEM55B phosphorylation sites is PXD046394 (http://www.ebi.ac.uk/pride/archive/projects/PXD046394).

Unique materials used are readily available from the authors. Source data are provided with this paper.

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

## Acknowledgements

This work was supported by the Intramural Research Program of the NIH, National Heart, Lung, and Blood Institute (NHLBI) (ZIA HL006075, R.P.). We thank Shawn Burgess (NHGRI, NIH) for allowing us to use the Applied Biosystems ABI 3730xl sequencer. We also thank Drs. Philip McCoy, Maria-Lopez Ocasio, Wan-Chi Lin and Pradeep Dagur (Flow Cytometry Core, NHLBI) for their assistance in the Annexin V apoptosis and ROS levels analysis. Graphics in Figs. 1a, 8d and Supplementary Fig. 1b were created with BioRender.com.

## Author contributions

E.J., R.W., A.R. and M.L.S. were involved in the experimental strategy, performed the experiments, analyzed the data, and participated in the preparation of the manuscript; R.P. designed the research, analyzed the data, supervised the project, and wrote the manuscript. All the authors reviewed the manuscript.

## Funding

## Competing interests

The authors declare no competing interest.
