## [Peer Review File · Nature Communications]

TMEM55B links autophagy flux, lysosomal repair, and TFE3 activation in response to oxidative stressREVIEWER COMMENTS

Reviewer #1 (Remarks to the Author):

Summary:

This is an interesting manuscript, where Jeong and colleagues studied the role of the lysosomal transmembrane TMEM55B under oxidative stress conditions using NaAsO₂. They showed that TMEM55B is interacting with NEDD4-like E3 ligases via its PPXY motif leading to their increased recruitment during oxidative stress. This leads to ubiquitylation of TMEM55B as well as interacting protein PLEKHM1 causing proteasomal degradation of PLEKHM1 and blocked autophagosome/lysosome fusion. Furthermore, they could convincingly prove that TMEM55B promotes recruitment of components of the ESCRT machinery during stress via its P(S/T)AP domain and sequesters FLCN/FNIP1 at the lysosome and thereby enables TFE3 translocation into the nucleus. Lastly, they showed in a zebrafish model that TMEM55B loss leads to less fitness upon arsenite treatment and earlier death. The manuscript was clear, and data nicely processed. Overall, I approve publishing of this work, if my comments below are addressed.

Major comments:

Figure 1h/3a/3b/4a:

The immunoprecipitation explained in the methods section, which refers to Fig. 1h, was not performed under denaturing conditions (Urea or high SDS content). Thus, ubiquitin chains/smears that are detected do not necessarily have to be linked to the pulled down protein of interest but could be also a result of ubiquitylated interactome. Please repeat all IPs, which aim for proving ubiquitylation of a particular protein, in denaturing conditions to remove non-covalent interactions.

In Fig. 4a ubiquitylation was not even shown with an anti-ubiquitin antibody. Please add in all denaturing IPs mentioning ubiquitination anti-ubiquitin antibody panels.

In this context, it would be also interesting to provide more information about the ubiquitin linkage type of TMEM55B and the particular lysines necessary for ubiquitylation by K->R mutations. Is ubiquitylation necessary for the interaction with PLEKHM1 and causative for disrupted JIP4 binding? Please provide data of TMEM55B-PLEKHM1/JIP4 interaction with TMEM55B K->R mutations and/or E1 inhibitors to block ubiquitylation.

Fig. 5e/5f:

To draw the conclusion that Galectin-3 punctae accumulate due to less recruitment of ESCRT machinery and therefore failed repair/more severe damage, one must compare colocalization events between Gal3 and ESCRT candidates. Here one would expect less colocalization events between Gal3 and ESCRT as it is suggested from your previous data. Only counting Gal3 dots leads to not interpretable results, since it is known that Gal3 also acts as upstream recruiting factor of ESCRTs. Thus, more punctae could simply also mean more recruitment of ESCRTs via this route as counteracting factor to compensate for TMEM55B loss.

Furthermore, it is mentioned in the discussion that TMEM55B depletion did not affect lysophagy. This was not studied at all in this manuscript. Only Gal3 punctae after 6h washout were compared, which is too early to study lysophagy. One needs more data to clearly distinguish between lysophagy and repair. Please remove this statement or include more data.

In the discussion section, the authors mentioned that NaAsO₂ causes TMEM55B phosphorylation. However, the authors do not provide any data underlining this statement besides the size shift on WB, which they speculatively call phosphorylation (line 109). So far, the data are missing to add phosphorylation in this pathway. Please provide either a phosphospecific TMEM55B antibody, phosphomimetic mutants or kinase inhibitors to confirm your hypothesis.

Minor comments:

Fig. 1d:

In the text it is mentioned that overexpression of NEDD4 alone leads to a diffuse cytosolic distribution. However, Figure 1d only shows co-expressions. Please add the data you are referring to.

Extended Data Figure 2f:

All other IPs showed and the text mentions that TMEM55B and PLEKHM1 only interact under stress condition. This IP is showing interaction without treatment. Is this a labeling mistake and these conditions were all treated? Where is that inconsistency coming from?

Fig. 4g: LC3 II level blot is not interpretable. Please repeat and quantify.

Extended Data Figure 5b:

Please add a panel for TMEM55B SA41 mutant showing that ubiquitylation is functional and colocalizing with TMEM55B, but not sufficient for CHMP2B recruitment.

Reviewer #2 (Remarks to the Author):

The manuscript by Jeong and colleagues addresses novel roles of TMEM55B in coordinating autophagy flux, lysosomal repair and TFE3 activation specifically in response to oxidative stress.

TMEM55B has been previously implicated in lysosomal positioning, particularly under starvation conditions (including some work from the same lab as this manuscript). Here, the authors opened a novel page in the roles played by TMEM55B, by showing that it regulates lysosomal repair (after lysosomal damage caused by oxidative stress) and sequesters the autophagosomes away from the damaged lysosomes.

First, the authors looked for novel binding partners of TMEM55B, and identified the NEDD4L family. Next, through a series of elegant experiments, they showed that there was extended binding of TMEM55B to NEDD4L under sodium arsenite treatment (and not for example under H₂O₂, amino acid starvation or ER stress). Next, the authors showed that TMEM55B is ubiquitinated by NEDD4L family members under arsenite treatment, and the interaction of TMEM55B with the protein PLEKHM1 increases. Given the interaction TMEM55B-NEDD4L, the proximity of NEDD4L results in the ubiquitination of PLEKHM1 and its proteasomal degradation, thus limiting these lysosomes from fusing with autophagosomes. Furthermore, TMEM55B promotes the recruitment of ESCRT complex proteins, which facilitate the repair of damaged lysosomal membranes, also specifically under sodium arsenite treatment. TMEM55B also sequesters FLCN, which eventually results in the release of TFE3 to activate transcription. The authors showed that the pathway is operating also *in vivo*, using zebrafish models in which they deleted the two fish homologues of TMEM55B.

Overall the manuscript is very good, the experiments well-designed are performed carefully and the data are clean. The authors generated point mutants of TMEM55B that specifically interact with NEDD4L and PLEKHM1, which yielded very elegant experiments. I have a

couple of small details that are mentioned further below.

I do however see one particular aspect of this otherwise excellent study that falls short: it remains unclear what arsenite does that damages lysosomes and triggers what the authors often coin as a response specific to oxidative stress. H₂O₂ also induces oxidative stress, in principle, but it does not trigger the same kind of effects observed with sodium arsenite. I imagine that this may be due to the fact that arsenite generates a lot more superoxide, but the only way to know is that the authors provide this information. It would be highly beneficial for the understanding of this pathway if the authors do a basic characterization of the redox parameters of these cells when they are treated with sodium arsenite or with H₂O₂, just to mention those two that are directly linked with redox metabolism. Furthermore, if the effects of arsenite are due to superoxide, then they should be mimicked by depletion of SOD1 (if the superoxide levels are cytoplasmic) or SOD2 (if the superoxide culprit is in the mitochondrial matrix). Would the silencing of any of these proteins result in lysosomal damage that activates TMEM55B and the downstream pathway here described? This would be an important experiment. Would depletion of GSH levels have a similar effect? I believe that to make a statement that this pathway responds to oxidative stress, the authors ought to measure important aspects of redox homeostasis.

Minor details:

- Fig 1f should be quantified

Reviewer #3 (Remarks to the Author):

This manuscript reports work suggesting that the lysosomal transmembrane protein TMEM55B orchestrates cellular responses to acute oxidative stress acting as a molecular sensor to coordinate autophagosome degradation, lysosomal repair and activation of stress responses. It follows previous work from Dr Puertollano's group, which previously demonstrated that TMEM55B binds to the dynein adaptor JIP4 to promote lysosomal transport to the perinuclear region and showed that following starvation, activation of TFEB/TFE3 increases TMEM55B levels and thus JIP4 recruitment to lysosomes, thereby facilitating autophagosome-lysosome fusion and hence being critical to coordinating cellular adaptation to nutrient deprivation.

The starting point of the work described in the present manuscript was the identification of several NEDD4-like E3 ligases as interactors of TMEM55B and, by using a P66A mutant, the demonstration that the PPXY motif in TMEM55B is responsible for ligase binding, although not required for binding to JIP4. The authors then conducted a series of further pull down/IP experiments identifying, confirming and analysing proteins, which showed increased binding to TMEM55B in cells subjected to acute oxidative stress by incubation with NaAsO₂. First, they observed increased binding of NEDD4L to TMEM55B upon NaAsO₂ treatment and also increased binding of the NEDD4L target PLEKHM1, whereas JIP4 binding was abolished. Interestingly, PLEKHM1 ubiquitination increased upon NaAsO₂ treatment, leading to proteasome-mediated degradation of PLEKHM1, accompanied by a decrease in autophagy flux, which did not occur in cells expressing TMEM55B P66A. Second, they found that NaAsO₂ treatment increased binding of several ESCRT proteins to TMEM55B, (dependent on both its PPXY and P(S/T)AP motifs), which they linked to ESCRT-dependent membrane repair of injured lysosomes, specifically under conditions of oxidative stress. Third, they showed increased binding of FLCN and FNIP1 to TMEM55B in NaAsO₂ treated

cells, which they linked to non-canonical activation of TFE3 in response to oxidative stress. In experiments linking their findings to the physiological effects of oxidative stress, the authors showed that TMEM55B-KO zebrafish embryos had a lower survival rate than WT embryos when exposed to prolonged NaAsO₂ treatment.

Overall, this manuscript provides a compelling case for the importance of TMEM55B in coordinating a cell's response to acute oxidative stress and certainly identifies binding partners of TMEM55B, whose binding changes upon to NaAsO₂ treatment. However, what is less certain is the relative importance of the different pathways outlined, exactly how these contribute to a cell's overall response to NaAsO₂ treatment and how general the response is to other agents causing acute oxidative stress. There are several major points, which the authors need to address:

Major points

1. TMEM55B was originally described as a PI4,5P2 4-phosphatase (doi: 10.1073/pnas.0509740102). This is not mentioned in the manuscript. However, if TMEM55B is so important to a cell or organism's response to NaAsO₂ treatment, to what extent is this a feature of effects on its enzymatic activity. Do the changes in phosphorylation or ubiquitination reported in this paper affect the enzymatic activity and overall cellular phosphoinositide composition/metabolism ?
2. Many of the pull down/IP blots shown are representative of 3 experiments, yet no attempt is made to quantitate the effect of treatments/mutations by measuring the density of blot bands (with an appropriate loading control - presumably the GFP band). This makes it difficult to get a sense of reproducibility between experiments. The data from different figures suggests that sometimes there is considerable variation eg the pull down in Fig1f shows a clear effect of NaAsO₂ treatment to increase NEDD4L bound to TMEM55B, yet in Fig5c the same conditions show barely any increase in NEDD4L bound to the WT TMEM55B.
3. In several of the pull down/IP experiments in which binding partners were identified, the logic of why one binding partner was followed up at the expense of others is not clear. Thus, why having characterized NEDD4 binding to TMEM55B so thoroughly at the start of the Results section was NEDD4L selected for follow-up after NaAsO₂ treatment when NEDD4 binding is equally dependent on the PPXY motif in TMEM55B of NaAsO₂ treated cells (see Fig5a) ? Also, why was PLEKHM1 followed up as a TMEM55B-binding NEDD4L target rather than VPS41 ? Do the authors have any comments on the relative importance of PLEKHM1 and VPS41 binding to TMEM55B in oxidative stress ?
4. On page 13 L253, the authors suggest that TMEM55B -mediated PLEKHM1 degradation may serve as a mechanism to slow down the delivery of degradative cargo when lysosomes undergo damage and described the reduction in PLEKHM1 levels in TMEM55B-GFP-expressing cells accompanying a decrease in autophagy flux. What is the effect on autophagy/autophagy flux of NaAsO₂ treatment of cells used in this study and how does it relate to the findings in the TMEM55B-GFP-expressing cells. In other cell types, NaAsO₂ treatment (albeit for longer times than applied to the cells used in the present study) has been described as inducing autophagy (see eg doi:10.15436/2575-808X.17.1470; doi: 10.1177/09603271221149196).
5. P18 L379 and Fig 8c. Much of the paper concerns acute oxidative stress in cultured cells treated with 300microM NaAsO₂ for 2h, but the zebra fish embryos were treated with 2mM NaAsO₂ for up to 7 days. Do the authors have any data on changes in the WT and KO embryos after acute exposure to NaAsO₂ eg changes in autophagy ?

Minor Points

1. P5 L84. TMEM55B-GFP co-IPs endogenous NEDD4. Does IPing endogenous TMEM55B co-IP NEDD4 as it does NEDD4L (Fig 2b) ?
2. P5 L90. There should really be a reference to Fig 5a here because it is not just NEDD4

binding that is dependent on the PPXY motif but also several other NEDD4-like ligases.

3. P6 L102 and Fig1e. Is there any significance to the ITCH band that remains present after pull down with the P66A mutant of TMEM55B-GFP ?

4. P6 L113, Extended data Fig 1d. The higher efficiency of pull-down under stress conditions should be quantified (with reference to a loading control) as it is not very obvious from the figure.

5. P7 L115/116 and Extended data Fig 1e. The figure shows reduction rather than abolition of the interaction of NEDD4L with TMEM55B-GFP P66A under stress conditions.

6. P7 L136. Other NEDD4-like E3 ligases are not shown in Extended data Fig 1f and 1g.

7. P7 L138 and Fig1g. Co-localization is stated and shown in the images. This should be backed up by Pearson's and/or Manders' colocalization coefficients for several cells.

8. P16 L338 and Figs 7c and 7d. The authors need to explain why, in a population of TMEM55B KO cells (clonal ?), there are still ~30% of cells showing FLCN recruitment to lysosomes after NaAsO₂ treatment. Is this significantly higher than control ?

9. Fig5a. The volcano plot presented is for hits identified in IPs of WT and P66A TMEM55B-GFP from cells treated with NaAsO₂. Do the authors have a data set for untreated cells and how does the volcano plot differ?

RESPONSE TO REVIEWERS

Reviewer #1 (Remarks to the Author):

Summary:

This is an interesting manuscript, where Jeong and colleagues studied the role of the lysosomal transmembrane TMEM55B under oxidative stress conditions using NaAsO₂. They showed that TMEM55B is interacting with NEDD4-like E3 ligases via its PPXY motif leading to their increased recruitment during oxidative stress. This leads to ubiquitylation of TMEM55B as well as interacting protein PLEKHM1 causing proteasomal degradation of PLEKHM1 and blocked autophagosome/lysosome fusion. Furthermore, they could convincingly prove that TMEM55B promotes recruitment of components of the ESCRT machinery during stress via its P(S/T)AP domain and sequesters FLCN/FNIP1 at the lysosome and thereby enables TFE3 translocation into the nucleus. Lastly, they showed in a zebrafish model that TMEM55B loss leads to less fitness upon arsenite treatment and earlier death. The manuscript was clear, and data nicely processed. Overall, I approve publishing of this work, if my comments below are addressed.

We thank the reviewer for the support.

Major comments:

Figure 1h/3a/3b/4a:

The immunoprecipitation explained in the methods section, which refers to Fig. 1h, was not performed under denaturing conditions (Urea or high SDS content). Thus, ubiquitin chains/smears that are detected do not necessarily have to be linked to the pulled down protein of interest but could be also a result of ubiquitylated interactome. Please repeat all IPs, which aim for proving ubiquitylation of a particular protein, in denaturing conditions to remove non-covalent interactions. In Fig. 4a ubiquitylation was not even shown with an anti-ubiquitin antibody. Please add in all denaturing IPs mentioning ubiquitination anti-ubiquitin antibody panels.

The reviewer raises an important point. As suggested, we have now repeated the indicated immunoprecipitations under denaturing conditions. Immunoprecipitation of TMEM55B-WT and TMEM55B-P66A under denaturing conditions is now shown in a new panel in **Figure 1h**. Please note that we confirmed increased ubiquitination of TMEM55B-WT, but not the P66A mutant, following treatment with NaAsO₂. Likewise, ubiquitination of PLEKHM1 under denaturing conditions is now shown as **Figure 3f**.

Please also note that we previously used mass spectrometry (MS) to confirm TMEM55B ubiquitination, allowing us to detect increased ubiquitination of K96, K103, K114, K120, K121, K134, and K148 in response to oxidative stress (**Supplementary Table 2**).

In this context, it would be also interesting to provide more information about the ubiquitin linkage type of TMEM55B and the particular lysines necessary for ubiquitylation by K->R mutations. Is ubiquitylation necessary for the interaction with PLEKHM1 and causative for disrupted JIP4 binding? Please provide data of TMEM55B-PLEKHM1/JIP4 interaction with TMEM55B K->R mutations and/or E1 inhibitors to block ubiquitylation.

We thank the reviewer for the suggestion. We now show that treatment with the E1 inhibitor TAK243 does not affect binding of TMEM55B to PLEKHM1 in response to NaAsO₂, nor the dissociation of the TMEM55B/JIP4 complex, suggesting that ubiquitination is not required for these interactions (**Supplementary Figure 4a**). This is consistent with **Figure 4a**, showing

efficient interaction of the TMEM55B-P66A mutant with PLEKHM1. Importantly, TAK243 prevented NaAsO₂-induced PLEKHM1 degradation (**Supplementary Figure 4a and 4b**), further confirming that this is a ubiquitination-mediated process. Incubation with TAK243 blocked the global increase in protein ubiquitination observed in total cell lysates after NaAsO₂ treatment, validating the efficiency of the compound (**Supplementary Figure 4a**).

Fig. 5e/5f:

To draw the conclusion that Galectin-3 punctae accumulate due to less recruitment of ESCRT machinery and therefore failed repair/more severe damage, one must compare colocalization events between Gal3 and ESCRT candidates. Here one would expect less colocalization events between Gal3 and ESCRT as it is suggested from your previous data. Only counting Gal3 dots leads to not interpretable results, since it is known that Gal3 also acts as upstream recruiting factor of ESCRTs. Thus, more punctae could simply also mean more recruitment of ESCRTs via this route as counteracting factor to compensate for TMEM55B loss.

While we clearly see recruitment of endogenous ESCRTs to lysosomes in cells expressing recombinant TMEM55B, it is very difficult to detect recruitment in non-transfected cells (please keep in mind that most studies use over-expression of recombinant ESCRTs for these type of experiments), and this prevented us from assessing Gal3/ESCRT co-localization in TMEM55B-depleted cells. However, the fact that we see increased cell death in TMEM55B-KO cells upon stress, strongly suggests that the increased number of Gal3 puncta is likely not a consequence of a compensatory increase in ESCRT recruitment.

Furthermore, it is mentioned in the discussion that TMEM55B depletion did not affect lysophagy. This was not studied at all in this manuscript. Only Gal3 punctae after 6h washout were compared, which is too early to study lysophagy. One needs more data to clearly distinguish between lysophagy and repair. Please remove this statement or include more data.

We agree with the reviewer and, as suggested, we have now removed this statement.

In the discussion section, the authors mentioned that NaAsO₂ causes TMEM55B phosphorylation. However, the authors do not provide any data underlining this statement besides the size shift on WB, which they speculatively call phosphorylation (line 109). So far, the data are missing to add phosphorylation in this pathway. Please provide either a phosphospecific TMEM55B antibody, phosphomimetic mutants or kinase inhibitors to confirm your hypothesis.

To confirm that TMEM55B undergoes phosphorylation in response to NaAsO₂ we have now performed two additional set of experiments. First, lysates and TMEM55B immunoprecipitates obtained from cells treated with NaAsO₂ were incubated with or without Lambda phosphatase. As predicted, the TMEM55B higher molecular weight band observed upon NaAsO₂ treatment almost disappeared after incubation with by Lambda phosphatase, strongly suggesting that this modification does indeed correspond to phosphorylation (**Figure 3a**). Quantification of three independent experiments is shown in **Figure 3b**. Second, we performed mass spectrometry analysis with the goal to identify specific changes in TMEM55B phosphorylation in response to oxidative stress. As seen in **Supplementary Figure 3a and Supplementary Tables 3 and 4**, we identified two TMEM55B residues, T111 and S162, which undergo phosphorylation in response to NaAsO₂. The percentage of TMEM55B peptides showing phosphorylation at T111 raised from 0.04% to 1.66%. Although this increase represents an over 40-fold increase, the number of phosphorylated peptides (1.66%) is still relatively low. In contrast, the percentage of peptides phosphorylated at S162 increased from 6.98% to almost 30% in NaAsO₂-treated cells, suggesting

that this residue may play an important function in TMEM55B regulation. As we showed in the previous version of the manuscript, inhibition of several key kinases, including mTOR, Erk1/2, JNK, or p38 MAPK, did not prevent TMEM55B phosphorylation or JIP4 dissociation (**Supplementary Figure 3d and 3e**). Our future studies will focus on trying to identify the kinase implicated in this regulation.

Minor comments:

Fig. 1d:

In the text it is mentioned that overexpression of NEDD4 alone leads to a diffuse cytosolic distribution. However, Figure 1d only shows co-expressions. Please add the data you are referring to.

Figure 1d now includes additional panels showing the distribution of mCherry-NEDD4 in cells infected with Ad-GFP, Ad-TMEM55B-WT and Ad-TMEM55B-P66A.

Extended Data Figure 2f:

All other IPs showed and the text mentions that TMEM55B and PLEKHM1 only interact under stress condition. This IP is showing interaction without treatment. Is this a labeling mistake and these conditions were all treated? Where is that inconsistency coming from?

We have now repeated the experiment shown in **Supplementary Figure 2f**, adding conditions with or without NaAsO₂ treatment. These data confirmed a robust increase in the binding of PLEKHM1-Flag to both, TMEM55B-full length (FL) and TMEM55B cytosolic domain (CD). We do detect some minor residual binding between PLEKHM1-Flag and TMEM55B-CD in untreated cells, which is likely due to the very high expression levels achieved when using recombinant proteins. However, the interaction is clearly increased by NaAsO₂, as expected.

Fig. 4g: LC3 II level blot is not interpretable. Please repeat and quantify.

As requested, the indicated experiment has been repeated and quantified (**Figure 4g and 4h**)

Extended Data Figure 5b:

Please add a panel for TMEM55B SA41 mutant showing that ubiquitylation is functional and colocalizing with TMEM55B, but not sufficient for CHMP2B recruitment.

As requested by the reviewer, we have now repeated the experiment shown in **Supplementary Figure 5b** to include distribution of TMEM55B-S41A, CHMP2B, and ubiquitin. Also, please note that in the previous version of the manuscript we had shown co-localization of TMEM55B-S41A with ubiquitin in **Supplementary Figure 5a**, as well as the inability of the mutant to recruit CHMP2B in **Figure 5d**.

Reviewer #2 (Remarks to the Author):

The manuscript by Jeong and colleagues addresses novel roles of TMEM55B in coordinating autophagy flux, lysosomal repair and TFE3 activation specifically in response to oxidative stress. TMEM55B has been previously implicated in lysosomal positioning, particularly under starvation conditions (including some work from the same lab as this manuscript). Here, the authors opened a novel page in the roles played by TMEM55B, by showing that it regulates lysosomal repair (after lysosomal damage caused by oxidative stress) and sequesters the autophagosomes away from the damaged lysosomes. First, the authors looked for novel

binding partners of TMEM55B, and identified the NEDD4L family. Next, through a series of elegant experiments, they showed that there was extended binding of TMEM55B to NEDD4L under sodium arsenite treatment (and not for example under H₂O₂, amino acid starvation or ER stress). Next, the authors showed that TMEM55B is ubiquitinated by NEDD4L family members under arsenite treatment, and the interaction of TMEM55B with the protein PLEKHM1 increases. Given the interaction TMEM55B-NEDD4L, the proximity of NEDD4L results in the ubiquitination of PLEKHM1 and its proteasomal degradation, thus limiting these lysosomes from fusing with autophagosomes. Furthermore, TMEM55B promotes the recruitment of ESCRT complex proteins, which facilitate the repair of damaged lysosomal membranes, also specifically under sodium arsenite treatment. TMEM55B also sequesters FLCN, which eventually results in the release of TFE3 to activate transcription. The authors showed that the pathway is operating also in vivo, using zebrafish models in which they deleted the two fish homologues of TMEM55B.

Overall the manuscript is very good, the experiments well-designed are performed carefully and the data are clean. The authors generated point mutants of TMEM55B that specifically interact with NEDD4L and PLEKHM1, which yielded very elegant experiments.

We thank the reviewer for the support.

I have a couple of small details that are mentioned further below.

I do however see one particular aspect of this otherwise excellent study that falls short: it remains unclear what arsenite does that damages lysosomes and triggers what the authors often coin as a response specific to oxidative stress. H₂O₂ also induces oxidative stress, in principle, but it does not trigger the same kind of effects observed with sodium arsenite. I imagine that this may be due to the fact that arsenite generates a lot more superoxide, but the only way to know is that the authors provide this information. It would be highly beneficial for the understanding of this pathway if the authors do a basic characterization of the redox parameters of these cells when they are treated with sodium arsenite or with H₂O₂, just to mention those two that are directly linked with redox metabolism.

The reviewer brings up a very interesting point, why only NaAsO₂, and not H₂O₂, causes TMEM55B phosphorylation, JIP4 dissociation, and PLEKHM1/FLCN interaction? As suggested by the reviewer, one possibility of that NaAsO₂ generates more reactive oxygen species, at least at the specific concentrations used in this study. To test this possibility, we performed flow cytometric analysis of ROS production. As seen in **Supplementary Figure 3b**, both NaAsO₂ and H₂O₂ were capable of increasing ROS production, but NaAsO₂ did so with higher efficiency. To further support this idea, we treated cells with different concentrations of acrolein and spermidine, two compound known to cause oxidative stress. Interestingly, we found that treatment with acrolein for 4h produced very high levels of ROS (**Supplementary Figure 3b**) and consistently, also induced TMEM55B phosphorylation (**Figure 3c**). In contrast, spermidine, which did not generate much oxidative stress, did not induce TMEM55B phosphorylation or JIP4 dissociation (**Supplementary Figure 3b and 3c**). Overall, these results support our suggestion that TMEM55B is important for cellular response to acute oxidative stress.

Furthermore, if the effects of arsenite are due to superoxide, then they should be mimicked by depletion of SOD1 (if the superoxide levels are cytoplasmic) or SOD2 (if the superoxide culprit is in the mitochondrial matrix). Would the silencing of any of these proteins result in lysosomal damage that activates TMEM55B and the downstream pathway here described? This would be an important experiment. Would depletion of GSH levels have a similar effect? I believe that to

make a statement that this pathway responds to oxidative stress, the authors ought to measure important aspects of redox homeostasis.

As requested by the reviewer, we depleted SOD1 and SOD2, alone or in combination, in U2OS cells. As seen in Appendix Figure 1A (attached to this rebuttal letter), SOD1/SOD2 depletion was not sufficient to induce TMEM55B phosphorylation, JIP4 dissociation, or TMEM55B/PLEKHM1 interaction (**Appendix Figure 1a and 1b**). One potential reason is that depletion of these proteins did not cause as much increase in ROS as treatment with NaAsO₂ (**Appendix Figure 1c**), suggesting that under these conditions, lysosomes may not be sufficiently damaged to activate this pathway.

Minor details:

- Fig 1f should be quantified

As requested, the quantification of Figure 1f is now shown as **Supplementary Figure 1c**, confirming a significant increase in binding of TMEM55B to NEDD4L under NaAsO₂ conditions.

Reviewer #3 (Remarks to the Author):

This manuscript reports work suggesting that the lysosomal transmembrane protein TMEM55B orchestrates cellular responses to acute oxidative stress acting as a molecular sensor to coordinate autophagosome degradation, lysosomal repair and activation of stress responses. It follows previous work from Dr Puertollano's group, which previously demonstrated that TMEM55B binds to the dynein adaptor JIP4 to promote lysosomal transport to the perinuclear region and showed that following starvation, activation of TFEB/TFE3 increases TMEM55B levels and thus JIP4 recruitment to lysosomes, thereby facilitating autophagosome-lysosome fusion and hence being critical to coordinating cellular adaptation to nutrient deprivation. The starting point of the work described in the present manuscript was the identification of several NEDD4-like E3 ligases as interactors of TMEM55B and, by using a P66A mutant, the demonstration that the PPXY motif in TMEM55B is responsible for ligase binding, although not required for binding to JIP4. The authors then conducted a series of further pull down/IP experiments identifying, confirming and analysing proteins, which showed increased binding to TMEM55B in cells subjected to acute oxidative stress by incubation with NaAsO₂. First, they observed increased binding of NEDD4L to TMEM55B upon NaAsO₂ treatment and also increased binding of the NEDD4L target PLEKHM1, whereas JIP4 binding was abolished. Interestingly, PLEKHM1 ubiquitination increased upon NaAsO₂ treatment, leading to proteasome-mediated degradation of PLEKHM1, accompanied by a decrease in autophagy flux, which did not occur in cells expressing TMEM55B P66A. Second, they found that NaAsO₂ treatment increased binding of several ESCRT proteins to TMEM55B, (dependent on both its PPXY and P(S/T)AP motifs), which they linked to ESCRT-dependent membrane repair of injured lysosomes, specifically under conditions of oxidative stress. Third, they showed increased binding of FLCN and FNIP1 to TMEM55B in NaAsO₂ treated cells, which they linked to non-canonical activation of TFE3 in response to oxidative stress. In experiments linking their findings to the physiological effects of oxidative stress, the authors showed that TMEM55B-KO zebrafish embryos had a lower survival rate than WT embryos when exposed to prolonged NaAsO₂ treatment.

Overall, this manuscript provides a compelling case for the importance of TMEM55B in

coordinating a cell's response to acute oxidative stress and certainly identifies binding partners of TMEM55B, whose binding changes upon to NaAsO₂ treatment.

We thank the reviewer for the support.

However, what is less certain is the relative importance of the different pathways outlined, exactly how these contribute to a cell's overall response to NaAsO₂ treatment and how general the response is to other agents causing acute oxidative stress.

Please see our response to reviewer 2. We now show that other agents, such as acrolein, which generates high ROS levels, can induce TMEM55B phosphorylation and JIP4 dissociation (**Figure 3c and Supplementary 3b and 3c**).

There are several major points, which the authors need to address:

Major points

1. TMEM55B was originally described as a PI4,5P₂ 4-phosphatase (doi: 10.1073/pnas.0509740102). This is not mentioned in the manuscript. However, if TMEM55B is so important to a cell or organism's response to NaAsO₂ treatment, to what extent is this a feature of effects on its enzymatic activity. Do the changes in phosphorylation or ubiquitination reported in this paper affect the enzymatic activity and overall cellular phosphoinositide composition/metabolism?

This is a very important point that we in fact addressed in our previous manuscript (doi: 10.1038/s41467-017-01871-z). In this study we determined that TMEM55B does not function as a bona fide PtdIns-4,5-P₂ 4-phosphatase. This conclusion was based on several findings. First, the TMEM55B Cx₅R motif, which is required for catalytic activity, differs from other well-established phosphatases in that the arginine is followed by an isoleucine residue, as opposed to a serine or threonine. Second, recruitment of TMEM55B to the plasma membrane (via rapamycin-induced dimerization with Lyn₁₁-FRB-iRFP) did not change PtdIns-4,5-P₂ levels, while recruitment of INPP5E to the cell surface, which was used as a positive control, induced a nearly complete loss of plasma membrane PtdIns-4,5-P₂. Third, we did not detect phosphatase activity in *in vitro* experiments measuring the ability of purified TMEM55B to hydrolyze PtdIns-4,5-P₂. Furthermore, we previously showed that mutation of the Cx₅R motif (GFP-TMEM55B-C133S) did not affect the ability of TMEM55B to interact with JIP4 or induce lysosomal clustering upon over-expression.

To further corroborate these data, we performed additional experiments addressing the ability of the TMEM55B-C133S mutant to interact with JIP4, NEDD4L, and PLEKHM1 following treatment with NaAsO₂. As seen in **Appendix Figure 2** (attached to this rebuttal letter), mutation of the Cx₅R motif did not affect TMEM55B's ability to interact with its binding partners in response to oxidative stress.

2. Many of the pull down/IP blots shown are representative of 3 experiments, yet no attempt is made to quantitate the effect of treatments/mutations by measuring the density of blot bands (with an appropriate loading control - presumably the GFP band). This makes it difficult to get a sense of reproducibility between experiments. The data from different figures suggests that sometimes there is considerable variation eg the pull down in Fig1f shows a clear effect of NaAsO₂ treatment to increase NEDD4L bound to TMEM55B, yet in Fig5c the same conditions show barely any increase in NEDD4L bound to the WT TMEM55B.

To address the reviewer's concern, we now show quantification of the interaction between TMEM55B and NEDD4L under different stress conditions (**Supplementary Figure 1c**). These data confirm significant increased binding between the two proteins in NaAsO₂-treated cells. Some other IPs are difficult to quantify since no interaction is observed in control conditions. However, our results are highly reproducible, and in all cases, we show not just interaction between recombinant, but also endogenous proteins. For example, binding of PLEKHM1 to either recombinant or endogenous TMEM55B is shown in seventeen different panels, supporting the reproducibility of our data. (**Figure 2b-e and g, Figure 4a and 4b, Supplementary Figure 2a and 2d-f, Supplementary Figure 3d and 3e, Supplementary Figure 4a and 4c, Appendix Figure 1b, and Appendix Figure 2**).

3. In several of the pull down/IP experiments in which binding partners were identified, the logic of why one binding partner was followed up at the expense of others is not clear. Thus, why having characterized NEDD4 binding to TMEM55B so thoroughly at the start of the Results section was NEDD4L selected for follow-up after NaAsO₂ treatment when NEDD4 binding is equally dependent on the PPXY motif in TMEM55B of NaAsO₂ treated cells (see Fig5a)? Also, why was PLEKHM1 followed up as a TMEM55B-binding NEDD4L target rather than VPS41? Do the authors have any comments on the relative importance of PLEKHM1 and VPS41 binding to TMEM55B in oxidative stress?

Our goal was to confirm the interaction of TMEM55B with different members of the NEDD4-like family, as we did in Figure 1. The availability of good antibodies that allow us to follow, not just total protein levels but also activation status, was the main reason for later focus on NEDD4L. In addition, the clear reduction in PLEKHM1 levels following treatment with NaAsO₂, together with the ability of TMEM55B to facilitate NEDD4-mediated ubiquitination, suggested that both processes might be related. Furthermore, it is well-established that PLEKHM1 interacts with VPS41 and other components of the HOPS complex, and this interaction is critical to facilitate fusion between autophagosomes and lysosomes (*doi: 10.1016/j.molcel.2014.11.006*). So, our prediction is that the HOPS complex is being pull-down via its interaction with PLEKHM1. In future studies we will address whether PLEKHM1 is indeed required for TMEM55B-VPS41 interaction, as well as the potential TMEM55B-mediated ubiquitination of the HOPS complex.

4. On page 13 L253, the authors suggest that TMEM55B -mediated PLEKHM1 degradation may serve as a mechanism to slow down the delivery of degradative cargo when lysosomes undergo damage and described the reduction in PLEKHM1 levels in TMEM55B-GFP-expressing cells accompanying a decrease in autophagy flux. What is the effect on autophagy/autophagy flux of NaAsO₂ treatment of cells used in this study and how does it relate to the findings in the TMEM55B-GFP-expressing cells. In other cell types, NaAsO₂ treatment (albeit for longer times than applied to the cells used in the present study) has been described as inducing autophagy (see eg *doi:10.15436/2575-808X.17.1470*; *doi: 10.1177/09603271221149196*).

In agreement with previous studies, we did observe autophagy induction in response to treatment with NaAsO₂ in U2OS cells (**Appendix Figure 3**, attached to this rebuttal letter). Expression of TMEM55B-WT did not prevent this induction, but reduced autophagy flux, as assessed by the lack of LC3_{II} increase upon incubation with bafilomycin. In contrast, a clear bafilomycin-induced increase in LC3_{II} levels was seen in untransfected cells, as well as in cells transfected with either Ad-GFP or Ad-TMEM55B-P66A (**Appendix Figure 3 and Figure 4h**).

5. P18 L379 and Fig 8c. Much of the paper concerns acute oxidative stress in cultured cells treated with 300microM NaAsO₂ for 2h, but the zebra fish embryos were treated with 2mM

NaAsO₂ for up to 7 days. Do the authors have any data on changes in the WT and KO embryos after acute exposure to NaAsO₂ eg changes in autophagy?

Multiple studies have addressed the toxic effects of an acute sodium arsenite treatment on embryonic and adult zebrafish over the last decade. In the former case, most of the toxicity studies started the sodium arsenite treatment very early during development (~4 hours post fertilization, hpf), well before the beginning of the formation of all the major organs ([doi: 10.1242/dmm.031575](https://doi.org/10.1242/dmm.031575); [doi: 10.1016/j.neulet.2022.137042](https://doi.org/10.1016/j.neulet.2022.137042); [doi: 10.3390/biom12121833](https://doi.org/10.3390/biom12121833); [doi: 10.1016/j.aquatox.2008.11.007](https://doi.org/10.1016/j.aquatox.2008.11.007); [doi: 10.1016/j.aquatox.2014.04.006](https://doi.org/10.1016/j.aquatox.2014.04.006); [doi: 10.1002/jat.4520](https://doi.org/10.1002/jat.4520)). In those studies, different ranges of doses were tested (from ~0.5 μ M to ~10 mM), and several dose-dependent effects were reported. For example, Li and colleagues observed that a 2 mM NaAsO₂ treatment from 4 hpf induced malformations in the 50% of the population at 2 days post fertilization ([doi: 10.1016/j.aquatox.2008.11.007](https://doi.org/10.1016/j.aquatox.2008.11.007)). More recently, Silva and colleagues showed that exposing zebrafish embryos from 5 until 72 hpf to ~1.5 mM sodium arsenite affects the expression of multiple genes in the oxidative stress and p53 pathways, inducing apoptosis in treated larvae ([doi: 10.1002/jat.4520](https://doi.org/10.1002/jat.4520)). In another study, Fuse and colleagues treated zebrafish embryos from 4 dpf to 9 dpf with different doses of NaAsO₂, from 0.125 mM to 8 mM, which represent the no-observed-effect-concentration dosage and the dose with the highest lethality, respectively. Moreover, they observed that embryos survival was dose-dependent and they then used 1 mM or 2 mM in further experiments ([doi: 10.1016/j.taap.2016.06.012](https://doi.org/10.1016/j.taap.2016.06.012)).

It appears evident that, compared to cellular studies, the *in vivo* analysis on zebrafish embryos requires in general a higher dosage. That can be easily explained by the ways that the different compounds can be adsorbed by the embryos. Aqueous, highly soluble compounds like sodium arsenite generally can pass through the chorion ([doi: 10.1021/acs.chemrestox.9b00335](https://doi.org/10.1021/acs.chemrestox.9b00335)) and are absorbed mainly through the skin and via swallowing at later larval stages when the zebrafish digestive system has been fully developed (~4-5 dpf) ([doi: 10.1038/s41467-022-28434-1](https://doi.org/10.1038/s41467-022-28434-1); [doi: 10.1210/en.2015-1039](https://doi.org/10.1210/en.2015-1039)). However, the actual exposure in the different tissues might be different to the medium concentration due to their different metabolic activity or accessibility ([doi: 10.1021/acs.chemrestox.9b00335](https://doi.org/10.1021/acs.chemrestox.9b00335)).

Based on the afore mentioned results and data available from literature, we decide to perform the NaAsO₂ acute treatments from 24 hpf, a stage when most of the organs already started their development although they are not fully functional ([doi: 10.1002/aja.1002030302](https://doi.org/10.1002/aja.1002030302)). In our experimental setup, doses below 2 mM did not seem to affect the overall survival of WT and TMEM55-KO embryos.

Minor Points

1. P5 L84. TMEM55B-GFP co-IPs endogenous NEDD4. Does IPing endogenous TMEM55B co-IP NEDD4 as it does NEDD4L (Fig 2b)?

As requested, we have now confirmed the interaction between endogenous TMEM55B and NEDD4 (**Appendix Figure 4**, attached to this rebuttal letter)

2. P5 L90. There should really be a reference to Fig 5a here because it is not just NEDD4 binding that is dependent on the PPXY motif but also several other NEDD4-like ligases.

The reviewer is correct, and we in fact show binding of TMEM55B to different NEDD4-like ligases in **Figure 1e**, as well as in **Supplementary Figure 1a**.

3. P6 L102 and Fig1e. Is there any significance to the ITCH band that remains present after pull down with the P66A mutant of TMEM55B-GFP?

The reviewer is correct in that we still observe a very minor interaction of ITCH with the TMEM55B-P66A mutant, and this is very reproducible between different experiments. However, upon quantification, we shown that the binding of TMEM55B-P66A to ITCH is still significantly reduced when compared with TMEM55B-WT (**Appendix Figure 5**, attached to this rebuttal letter).

4. P6 L113, Extended data Fig 1d. The higher efficiency of pull-down under stress conditions should be quantified (with reference to a loading control) as it is not very obvious from the figure.

The required quantification is now shown as **Supplementary Figure 1f**.

5. P7 L115/116 and Extended data Fig 1e. The figure shows reduction rather than abolition of the interaction of NEDD4L with TMEM55B-GFP P66A under stress conditions.

Please note that quantification of four independent experiments confirmed that the interaction of TMEM55B-P66A with NEDD4L in NaAsO₂-treated cells is very significantly diminished when compared with TMEM55B-WT. These data are now shown as **Supplementary Figure 1h**.

6. P7 L136. Other NEDD4-like E3 ligases are not shown in Extended data Fig 1f and 1g.

We thank the reviewer for the suggestion. We have now assessed ITCH activation in response to NaAsO₂. It was previously described that phosphorylation of ITCH at threonine 222 (T222) enhances its catalytic activity (doi.org/10.1073/pnas.0510664103). As expected, we observed a robust increase in T222 phosphorylation following incubation with NaAsO₂, indicating that arsenite may have a broad effect in the activation of NEDD4-like E3 ligases (**Supplementary Figure 1k and 1l**).

7. P7 L138 and Fig 1g. Co-localization is stated and shown in the images. This should be backed up by Pearson's and/or Manders' colocalization coefficients for several cells.

As requested by the reviewer, Pearson's coefficients assessing colocalization between TMEM55B and ubiquitin are shown in **Supplementary Figure 1m**.

8. P16 L338 and Figs 7c and 7d. The authors need to explain why, in a population of TMEM55B KO cells (clonal ?), there are still ~30% of cells showing FLCN recruitment to lysosomes after NaAsO₂ treatment. Is this significantly higher than control?

Please note that we have now performed the requested statistical analysis and found that the differences in the recruitment of FLCN to lysosomes in TMEM55B-KO cells under control and NaAsO₂ conditions are not significant. This is now indicated in a revised graphic shown in **Figure 7d**.

9. Fig 5a. The volcano plot presented is for hits identified in IPs of WT and P66A TMEM55B-GFP from cells treated with NaAsO₂. Do the authors have a data set for untreated cells and how does the volcano plot differ?

Unfortunately, our previous MS analysis did not include untreated P66A-expressing cells. To address the reviewer's comment, we performed a new MS analysis of untreated cells expressing Ad-GFP, Ad-TMEM55B-WT and Ad-TMEM55B-P66A. As expected, we did not detect interaction of either TMEM55B-WT or TMEM55B-P66A with PLEKHM1, FLCN, or ESCRT subunits under these conditions. Furthermore, we confirmed reduced interaction of TMEM55B-P66A with

NEDD4-like ligases, while the binding to JIP4 was comparable between TMEM55B-WT and TMEM55B-P66A (**Appendix Figure 6**, attached to this rebuttal letter).

We want to thank all the reviewers one more time for their constructive and valuable comments. All the points raised were quite useful and helped us to improve our manuscript.

Appendix Figure 1. Depletion of SOD1 and SOD2 does not induce an acute oxidative stress response. **a**, siRNA transfected U2OS cells were treated with or without NaAsO₂ (300 μ M) for 2 h. **b**, siRNA transfected U2OS cells were infected with adenovirus expressing TMEM55B-GFP and treated with or without NaAsO₂ (300 μ M) for 2 h. Cells were lysed and pulled down with GFP beads. **c**, Intracellular ROS detection by FACS analysis in siRNA transfected U2OS cells treated with or without NaAsO₂ (300 μ M) for 2 h.

Appendix Figure 2. Mutation of C133 does not affect interaction of TMEM55B with its binding partners. U2OS cells transfected with plasmids encoding TMEM55B-GFP-WT or TMEM55B-GFP-C133S were treated with NaAsO₂ (300 μM) for 2 h. Cells were lysed and pulled down with GFP beads.

Appendix Figure 3. TMEM55-WT over-expression reduces autophagy flux. U2OS cells infected with adenovirus expressing GFP, TMEM55B-GFP-WT or TMEM55B-GFP-P66A were incubated with NaAsO₂ (300 μM) alone or NaAsO₂ (300 μM) plus Bafilomycin A1 (200 nM) for indicated times.

Appendix Figure 4. Interaction of TMEM55B with endogenous NEDD4. U2OS cells treated with or without NaAsO₂ (300 μM) for 2 h were lysed and immunoprecipitated with anti-TMEM55B antibody.

Appendix Figure 5. Mutation of the TMEM55B PPXY motif significantly reduces its interaction with ITCH. Quantification of immunoblots shown in (figure 1e)

Accession	#AAs	Gene Symbol	summed PSMs	GFP PSMs	TMEM55B-WT PSMs	TMEM55B-P66A PSMs	GFP Abundance	TMEM55B-WT Abundance	TMEM55B-P66A Abundance
AMQ45836	238	GFP	694	335	185	174	14990583016	2754841450	4104404711
Q86T03	277	PP4P1	238		107	131		2736553286	3367989999
O60271	1321	JIP4	186		96	90		137260651	197189857
P46934	1319	NEDD4	55		55			64469949.67	
Q96PU5	975	NED4L	22	3	15	4	1098713.953	5986280.594	1780902.281
Q9H0M0	922	WWP1	8		8			7912128.688	
O00308	870	WWP2	4		4			880164.1875	

Appendix Figure 6. Mass spectrometry analysis showing decreased binding of the TMEM55B-P66A mutant to NEDD4-like E3 ligases under control conditions.

REVIEWERS' COMMENTS

Reviewer #1 (Remarks to the Author):

The authors have sufficiently answered all questions and concerns. No further revisions are requested.

Reviewer #2 (Remarks to the Author):

The authors addressed adequately my concerns.

Reviewer #3 (Remarks to the Author):

The authors have responded appropriately to all the matters I raised when reviewing the original submission. I have no further questions or comments.